# A modular and synthetic biosynthesis platform for de novo production of diverse halogenated tryptophan-derived molecules

Kevin B. Reed[1,4], Sierra M. Brooks [1,4], Jordan Wells [1], Kristin J. Blake[2], Minye Zhao[1], Kira Placido[1], Simon d'Oelsnitz [3], Adit Trivedi[1], Shruti Gadhiyar[1] & Hal S. Alper [1,3] ✉

Halogen-containing molecules are ubiquitous in modern society and present unique chemical possibilities. As a whole, de novo fermentation and synthetic pathway construction for these molecules remain relatively underexplored and could unlock molecules with exciting new applications in industries ranging from textiles to agrochemicals to pharmaceuticals. Here, we report a mix-and-match co-culture platform to de novo generate a large array of halogenated tryptophan derivatives in *Escherichia coli* from glucose. First, we engineer *E. coli* to produce between 300 and 700 mg/L of six different halogenated tryptophan precursors. Second, we harness the native promiscuity of multiple downstream enzymes to access unexplored regions of metabolism. Finally, through modular co-culture fermentations, we demonstrate a plug-and-play bioproduction platform, culminating in the generation of 26 distinct halogenated molecules produced de novo including precursors to prodrugs 4-chloro- and 4-bromo-kynurenine and new-to-nature halogenated beta carbolines.

The vast majority of biochemistry revolves around a finite number of elements on the periodic table. However, by coupling novel synthetic tools with the exploration of diverse microbial biochemical space, new pathways for the assimilation of more uncommon elements are emerging[1–3]. This expanded scope is moving metabolic engineering away from simply C–C, C–H, C–O, C–N, and C–S bonds and toward alternative chemical diversity, thus expanding the cellular chemical palette[1,4,5]. Among these alternative pathways of interest is the incorporation of halogen elements into microbial metabolism[6–10]. Halogenation chemistry has become rather ubiquitous in modern society with broad applications spanning pharmaceuticals[11–14], agrochemicals[15–17], and novel materials[18–21]. The critical incorporation of a halogen atom into compounds directly confers function as seen in examples such as the widely used polymer polyvinylchloride (PVC), the potent antibiotic chloramphenicol, and the

flame retardant brominated polystyrene (BPS)[22]. Organic synthesis of halogenated compounds, while well-established and understood, is marred by highly toxic chemicals[15], poor atom economy[23], and limited stereo-/regio-selectivity[24] that hinders targeted halogenation in complex molecules[25]. Furthermore, the requirement for enantiomerically pure end products often complicates separation and purification chemical synthesis of these compounds[26]. In contrast, biological synthesis routes can bypass many of these limitations and theoretically offer an effective and more environmentally friendly alternative for producing halogenated molecules at near ambient conditions.

In nature, halogenase enzymes generate precisely halogenated end products through a variety of reaction mechanisms[27]. To this end, an array of halogenases have been discovered and can be characterized into four main classes. These include, for example, members of the Fe(II)/

[1]McKetta Department of Chemical Engineering, The University of Texas at Austin, 200 E Dean Keeton St. Stop C0400, Austin, TX, USA. [2]Mass Spectrometry Facility, Department of Chemistry, The University of Texas at Austin, 105 E 24th Street, Austin, TX, USA. [3]Institute for Cellular and Molecular Biology, The University of Texas at Austin, 2500 Speedway Avenue, Austin, TX, USA. [4]These authors contributed equally: Kevin B. Reed, Sierra M. Brooks. ✉e-mail: halper@che.utexas.edu

alpha KG-dependent class of halogenases such as SyrB2[28], BesD and similar enzymes[29], and late-stage halogenase WelO5[30]. Other classes consist of the haloperoxidases and SAM-dependent halogenases[17]. Lastly, flavin-dependent halogenases constitute the final class, including RadH[31] and Rdc2[32,33], late-stage halogenase MalA[34], and, of particular interest for this study, the tryptophan halogenases such as RebH, PyrH, and Thal[35–37]. Among these enzymes, the tryptophan halogenases have been most extensively studied and engineered over the past few decades mainly in vitro and recently in vivo. More specifically, the varied regioselectivity of these enzymes offers halogenation across the 5-, 6-, or 7-positions of tryptophan with the 7-position most studied in large-scale efforts[27]. The first gram-scale production of 7-chloro-tryptophan from a tryptophan feed was demonstrated in vitro using cross-linked enzyme aggregates (CLEA's)[38]. Recent efforts in the engineering of *Corynebacterium glutamicum* have demonstrated the first de novo and in vivo gram-scale production of 7-bromo-tryptophan[39]. These studies provide promise for expanding bio-based production to alternative hosts and halogenation positions within tryptophan at a scale relevant for commercially viable industrial production.

The combinatorial biochemistry afforded by linking halogenases with downstream enzymes can access a diverse array of halogenated compounds. Tryptophan itself serves as a gateway to a plethora of interesting natural products ranging from small molecules like indole, kynurenines, quinones, and tryptamines to larger molecules like violacein, strictosidine, and beta-carbolines[40–43]. To this end, several recent reports describe the in vivo production of tryptophan-derived, halogenated products. For example, halogenated kynurenine derivates, especially 4-chloro-kynurenine, were generated in *Streptomyces coelicolur*[44], halogenated indolocarbazoles have been generated via combinatorial biosynthesis in *S. albus*[45], and halogenated tryptophan derivatives for subsequent transition metal catalysis applications[46]. New-to-nature halogenated alkaloids[47], indigoids[48], and auxins[49] have been created in planta. Additionally, halogenated quinolines and alkaloids were realized using yeast bioproduction platforms[50,51]. While these examples demonstrate advances in halogenated metabolism, they do not ubiquitously describe de novo microbial production of diverse halogenated tryptophan-derived compounds starting from simple sugar starting materials. This is an important consideration given the relatively expensive and low-water-soluble substrates often used in prior studies such as indole, tryptophan, or other halogenated precursor molecules, as well as need for de novo production in planta to rely on variable, seasonally-dependent crop yields[51].

In this work, we harness halogenases and downstream pathways to generate industrially attractive halogenated molecules in a safe and effective manner through metabolic engineering and synthetic co-cultures. In doing so, we establish a co-culture system that uses mix-and-match technology to afford differential downstream halogenation type and position in a manner that enables combinatorial pathway assembly for diverse halogenated molecules. Specifically, we showcase high-level production of halogenated tryptophan analogs and a collection of strains/pathways that can enable subsequent transformation of these precursors into desirable compounds. Through a synthetic, modular co-culture system, 26 distinct halogenated molecules are generated de novo from glucose, including new-to-nature beta carbolines, prodrug precursors to 4-chloro- and 4-bromo-kynurenine, plant hormone precursors, and other pharmaceutically relevant precursor molecules including tryptamines and indoles. Taken together, this work unlocks halogenated biochemistry by uniting concepts from both combinatorial chemistry and synthetic biology.

## Results

### Enabling halogenated tryptophan production in an *E. coli* platform

While de novo production of halogenated tryptophan has been reported in *C. glutamicum* for 7-Br-tryptophan and detectable quantities of 7-Cl-tryptophan[39,52], no studies have reported such production (for these particular halogenated forms or others) of close to gram-scale titers in *E. coli*. Initial efforts here evaluated the synthetic expression of halogenases and flavin reductase cofactor rebalance partners to enable production (Fig. 1). Specifically, production of halogenated tryptophan was evaluated using varying expression regimes for Th-Hal and flavin reductases Th-Fre and EcFre (Supplementary Fig. 1). These results reinforce previous literature that the native *E. coli* flavin reductase (EcFre) is not expressed highly enough to enable sufficient cofactor rebalance of FAD to $FADH_2$ for high-level production of halo-tryptophan using tryptophan halogenases[53]. Specifically, only minimal halogenated tryptophan (e.g., 40 µM of 6-chloro-tryptophan from 1 mM of tryptophan fed) was produced without overexpression of a flavin reductase (Supplementary Fig. 1). Expression of a heterologous, more thermostable flavin reductase (Th-Fre) resulted in higher production over EcFre and the null strain, reinforcing the importance of optimizing the cofactor balance for the halogenase reaction as noted in prior studies[36–38,46,47]. This system also allowed for the optimization of copy number and promoters driving expression of Th-Hal as a model halogenase (Supplementary Note 1).

### Establishing a panel of halo-tryptophan precursor production through halogenase selection

Using the expression platform above, we next evaluated the in vivo halogenation profile for a collection of halogenases comprised of at least two homologues capable of catalyzing each regioselective reaction at the 5, 6, or 7 positions of tryptophan including well-studied halogenases like Thal[36], RebH[37], PrnA[54], and PyrH[35] (Table 1). Tryptophan feeding assays were conducted (Fig. 1C) at varying temperatures to investigate the robustness of each enzyme and evaluation of chloro- and bromo-preferences (Supplementary Fig. 2).

A few initial observations can be made from the halogenase panel. First, the halogenase XsHal displayed robust in vivo production of both 5-chloro- and 5-bromo-tryptophan precursors. At all temperatures, XsHal performs significantly better than its counterpart PyrH, an enzyme that has a very low reported melting temperature of around 30 °C[55]. Furthermore, XsHal is reported to have a 2-fold increase in catalytic efficiency over PyrH[56] in vitro, yet exhibits multi-fold higher production in vivo at various temperatures. Second, the halogenase Thal shows the most consistent generation of both 6-chloro- and 6-bromo-tryptophan precursors at 30 °C, whereas Th-Hal, a reported halogenase from a thermophilic organism, shows the highest conversion of the tryptophan to 6-halo-tryptophan at 37 °C, with an evident preference for chloro- over bromo-addition. For the 7-tryptophan halogenases, RebH, which was previously used for other de novo halogenated molecule production in planta[47], provided the most consistent conversion of tryptophan into 7-bromo- and 7-chloro-tryptophan precursors at 30 °C, whereas PrnA could be used as a reliable halogenase at higher temperatures. Lastly, halogenation was almost universally restricted at 25 °C with the exception of XsHal that enabled high turnover even at this suboptimal operating temperature. Based on these results, halogenases XsHal, Thal, and RebH were selected based on their superior conversion at 30 °C and ability to collectively access multiple halogenation sites for both chlorine and bromine on tryptophan.

### Enabling de novo production of halo-tryptophan precursors through metabolic engineering

After characterizing the potential of *E. coli* to express functional halogenases and selecting a collection of functional enzymes, we employed a metabolic engineering approach to improve precursor availability and boost halo-tryptophan precursor production de novo. This effort focused broadly on the three goals of: (i) removal of degradation mechanisms, (ii) removal of feedback regulation, and (iii) overexpression of biosynthetic pathway enzymes (Fig. 2A).

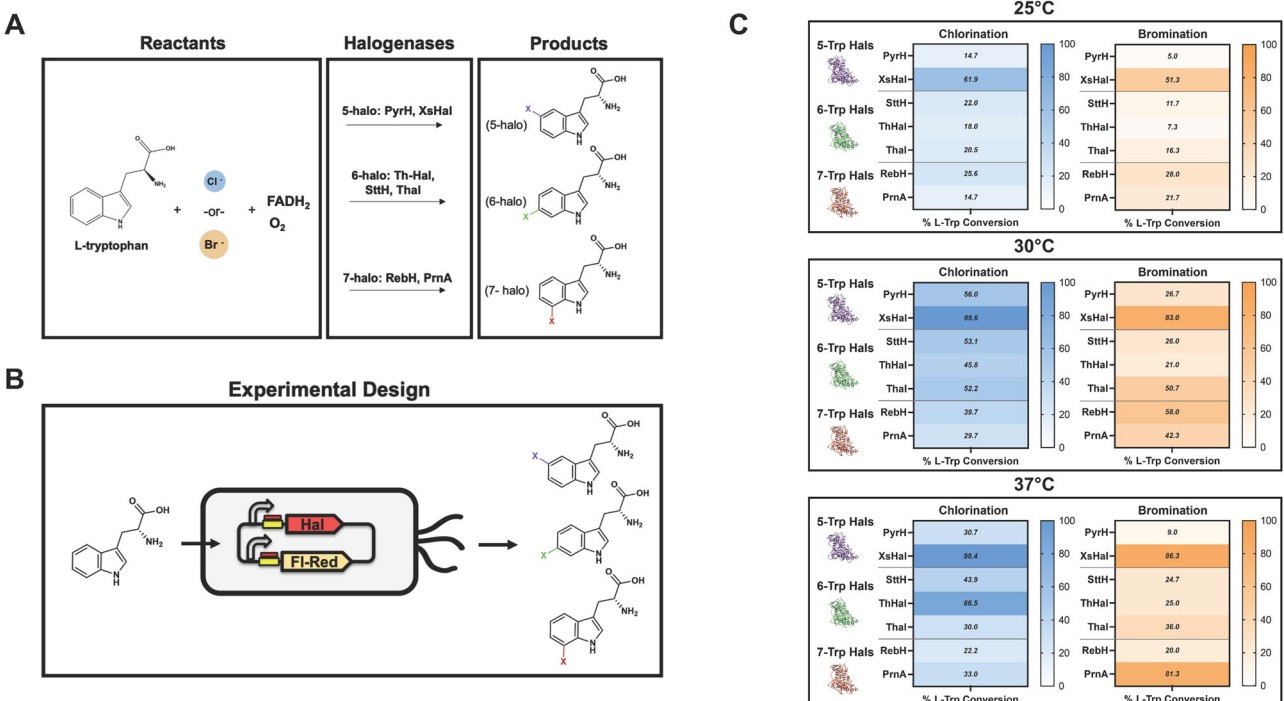

**Fig. 1 | Selection of halogenases to acquire a panel of modified precursors.**
**A** Reaction overview for tryptophan halogenases. 5-tryptophan-halogenases: PyrH, XsHal; 6-tryptophan-halogenases: Th-Hal, SttH, Thal; 7-tryptophan-halogenases: RebH, PrnA. **B** Experimental Design for halogenase panel. Strain sKR-160 containing respective halogenases and flavin reductase Th-Fre were supplied with 1 mM of tryptophan and production formation of respective halogenated tryptophan products (5-, 6-, 7-chloro-/bromo-tryptophan) was observed. **C** Panel of halogenases with calculated L-tryptophan conversion at varying temperatures: 25 °C, 30 °C, and 37 °C. Data are mean ± S.D.; *n* = 3 biological replicates. Source Data are provided as a Source Data file.

**Table 1 | List of halogenase enzymes used in this study**

| Halogenase | Organism of origin | Product formed | Halogenation position |
|---|---|---|---|
| PyrH | *Streptomyces rugosporus* | 5-halo-tryptophan | 5 |
| XszenFHal (XsHal) | *Xenorhabdus szentirmaii* | 5-halo-tryptophan | 5 |
| Thal | *Streptomyces albogriseolus* | 6-halo-tryptophan | 6 |
| Th-Hal | *Streptomyces violaceusniger* | 6-halo-tryptophan | 6 |
| SttH | *Streptomyces toxytricini* | 6-halo-tryptophan | 6 |
| RebH | *Lentzea aerocolonigenes* | 7-halo-tryptophan | 7 |
| PrnA | *Pseudomonas fluorescens* | 7-halo-tryptophan | 7 |

First, degradation was removed through targeting the tryptophanase (encoded by *tnaA*) for deletion to remove degradation into indoles[57] and deletion of the TrpR transcriptional repressor that serves to regulate biosynthesis and transport[58]. Disruption of the *trpR* and *tnaA* genes did not immediately show appreciable accumulation of tryptophan in the cellular supernatant after 24 h (Fig. 2B). Second, removal of feedback inhibition and overexpression of biosynthetic enzymes in the aromatic amino acid pathway were incorporated into the strain and leveraged efforts of many reports to improve tryptophan overproduction[59–61] including canonical targets including mutations in TrpE, AroG, and SerA to enable feedback resistance (fbr)[62]. A synthetic modularization of metabolism approach[63] was conducted here consisting of a Precursor Module and a Tryptophan Biosynthesis Module. In this case, the Precursor Module consisted of overexpressions of AroG(fbr) and SerA(fbr) and the Tryptophan Biosynthesis Module consists of all the genes in the trp operon with peptide leader TrpL removed and the feedback-resistant mutant of TrpE, TrpE(fbr). Expression strength and copy numbers of these modules were optimized to obtain high levels of tryptophan overproduction (Fig. 2B). Table 2 outlines the collective metabolic modifications investigated in this study. The final strain modification strategy (Supplementary Discussion 1) resulted in a strain (*E. coli* sKR-Trp4) capable of producing over 200 mg/L of tryptophan after 24 h in minimal media containing only 5 g/L glucose in a 96-deep well plate (1 mL scale) (Fig. 2B).

Using this high-tryptophan producing *E. coli* strain, it is possible to incorporate the above characterized halogenases to enable de novo production of halo-tryptophan precursors. To do so, the sKR-Trp4 strain was transformed with either XsHal, Thal, and RebH to generate three strains respectively (sKR-Trp4-XsHal, sKR-Trp4-Thal, and sKR-Trp4-RebH) (Fig. 3A). The resulting de novo titers of 0.3–0.7 g/L halogenated tryptophan for each corresponding halogenase fed with 40 g/L glucose and the corresponding halide salt are shown in Fig. 3B. These strains achieve a selectivity reaching as high as 96% for bromo-tryptophan even in the presence of competition for the halide salt with the residual chloride present in the growth media (Supplementary Fig. 3), consistent with previously reported selectivity in other organisms that use similar levels of bromide salt in the media[39]. Altogether, these findings showcase the highest titers of de novo varied halogenated tryptophan production at flask scale to date. It is important to

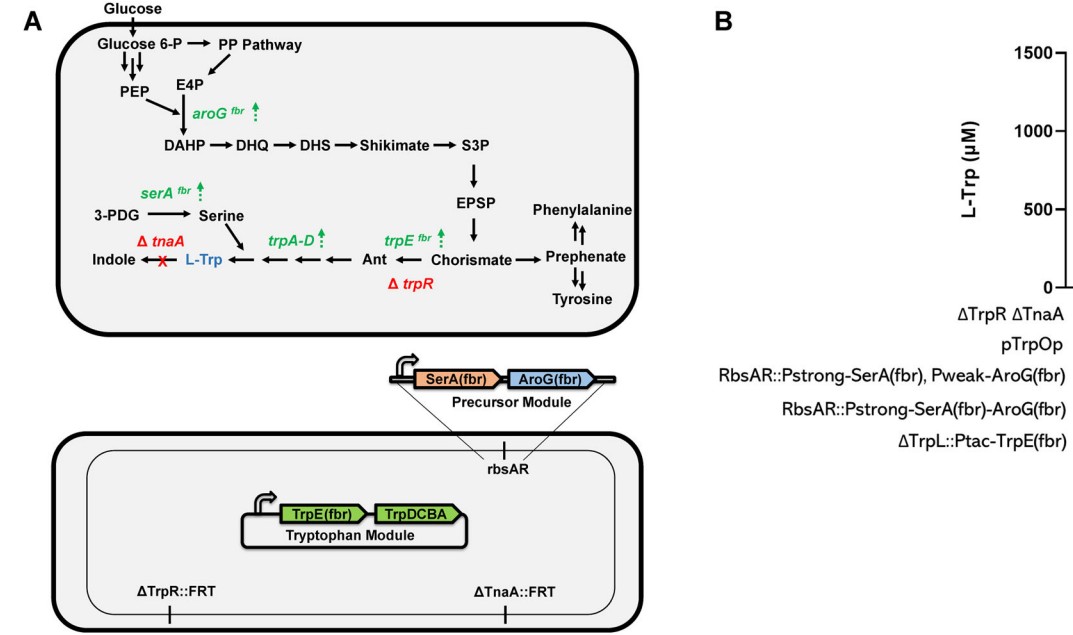

**Fig. 2 | Metabolic engineering in *E. coli* for L-tryptophan production. A** Genetic changes leading to high tryptophan production in *E. coli* BW25113 background and schematic of the tryptophan overproduction strategy for strain sKR-Trp4, the highest producer of tryptophan from glucose generated in this study. The outer loop within the bottom schematic of (**A**) represents the genome and relevant genomic modifications, including deletions of *TrpR* and *TnaA* and the integration of a precursor module containing *SerA(fbr)* and *AroG(fbr)* within the *rbsAR* locus. Glucose 6-P: glucose-6-phosphate; PP Pathway: pentose phosphate pathway; PEP: phosphoenolpyruvate; E4P: erythrose-4-phosphate; DAHP: 3-deoxy-D-arabino-heptulosonic acid 7-phosphate; DHQ: 3-dehydroquinate; DHS: 3-dehydroshikimate; S3P: shikimate-3-phosphate; EPSP: 5-enolpyruvylshikimate 3-phosphate; Ant: anthranilate; L-Trp: L-tryptophan; 3-PDG: 3-phospho-D-glycerate. **B** Tryptophan overproduction based on genetic modifications outlined in Table 2 and Supplementary Data 4. Each bar represents a biological triplicate for fermentations of 5 g/L glucose in minimal media after 24 h. Data are mean ± S.D.; *n* = 3 biological replicates. Source Data are provided as a Source Data file.

**Table 2 | Tryptophan overproduction strains with corresponding modifications constructed in this study**

| Strain name | Modifications |
| --- | --- |
| sKR-Trp0 | ΔTnaA::FRT |
| sKR-Trp1 | ΔTrpR::FRT and ΔTnaA::FRT |
| sKR-Trp2 | ΔTrpR::FRT, ΔTnaA::FRT, and pTrpOp-TrpE(fbr)DCBA |
| sKR-Trp3 | ΔTrpR::FRT, ΔTnaA::FRT, pTrpOp-TrpE(fbr)DCBA, and rbsAR::pJ23105-B32-SerA(fbr)-Pyibn-AroG(fbr) |
| sKR-Trp4 | ΔTrpR::FRT, ΔTnaA::FRT, pTrpOp-TrpE(fbr)DCBA, and rbsAR::pJ23105-B32-SerA(fbr)-B34-AroG(fbr) |
| sKR-Trp5 | ΔTrpR::FRT, ΔTnaA::FRT, ΔrbsAR::pJ23105-B32-SerA(fbr)-B34-AroG(fbr), and ΔtrpL::TP24-Ptac-TrpE(fbr)-TrpDCBA |

note that these products are secreted and thus enable an easy access point for creating a modular diversification approach using a co-culture.

**Initial halogen-product diversification through complementing promiscuous enzymes and feeding assays**
Tryptophan is the most chemically complex proteogenic amino acid and can quickly be modified in many positions through even just single enzymatic steps to yield products with a diverse range of applications (Fig. 4A). To enable an exploration of this chemical space with halogenated compounds, enzymes were selected to target as many biologically accessible reaction centers of tryptophan as possible (Fig. 4, as determined by Transform MinER online module[64]). A total of 10 modifying enzymes (beyond the halogenases described earlier) were selected and evaluated for their ability to convert 500 μM of fed L-tryptophan to a corresponding downstream product. Of the explored set of enzymes, five of these enzymes (encoded by RgnTDC, iaaM, TnaA, KynA, and McbB) enabled conversion of all fed tryptophan (Supplementary Data 1). Molecules corresponding to the theoretical product **2a**, **3a**, **4a**, **5a**, and **6a** were observed for each enzyme, respectively (Fig. 5A and Supplementary Figs. 4–41). Beyond this set,

TsrM was observed to catalyze the reaction of tryptophan to 2-methyl-L-tryptophan (**7a**), though showed very minimal conversion of tryptophan fed and yielded low intensities on the LC-MS samples (Supplementary Figs. 42–46). Other enzymes did not convert any appreciable tryptophan under various fermentation and expression conditions after 48 h and the corresponding theoretical products **8a**, **9a**, **10a**, and **11a** were not observed (Fig. 3b; shown in white in the "Product Observed" column). These enzymes comprise either prenyltransferases, including CymD[65], DmaW[66], and etpPT[67], and thus require a large pool of DMAPP to function well in vivo or are P450s, such as CYP79B2, with documented difficulties in soluble expression.

Nonetheless, a wide range of reactions are represented in this functional subset including a ring-opening, a ring-closing, cleavage of the amino acid group, and multiple modifications on the amino acid portion of tryptophan. Specifically, RgnTDC is a tryptophan decarboxylase (TDC) from the organism *Ruminococcus gnavus*, and catalyzes the formation of tryptamine, a physiologically important and highly relevant pharmaceutical precursor[68,69]. RgnTDC was previously characterized to be highly promiscuous towards many tryptophan derivatives, including halogenated ones[70]. iaaM is an enzyme involved in the production of auxin in plants, catalyzing the generation of

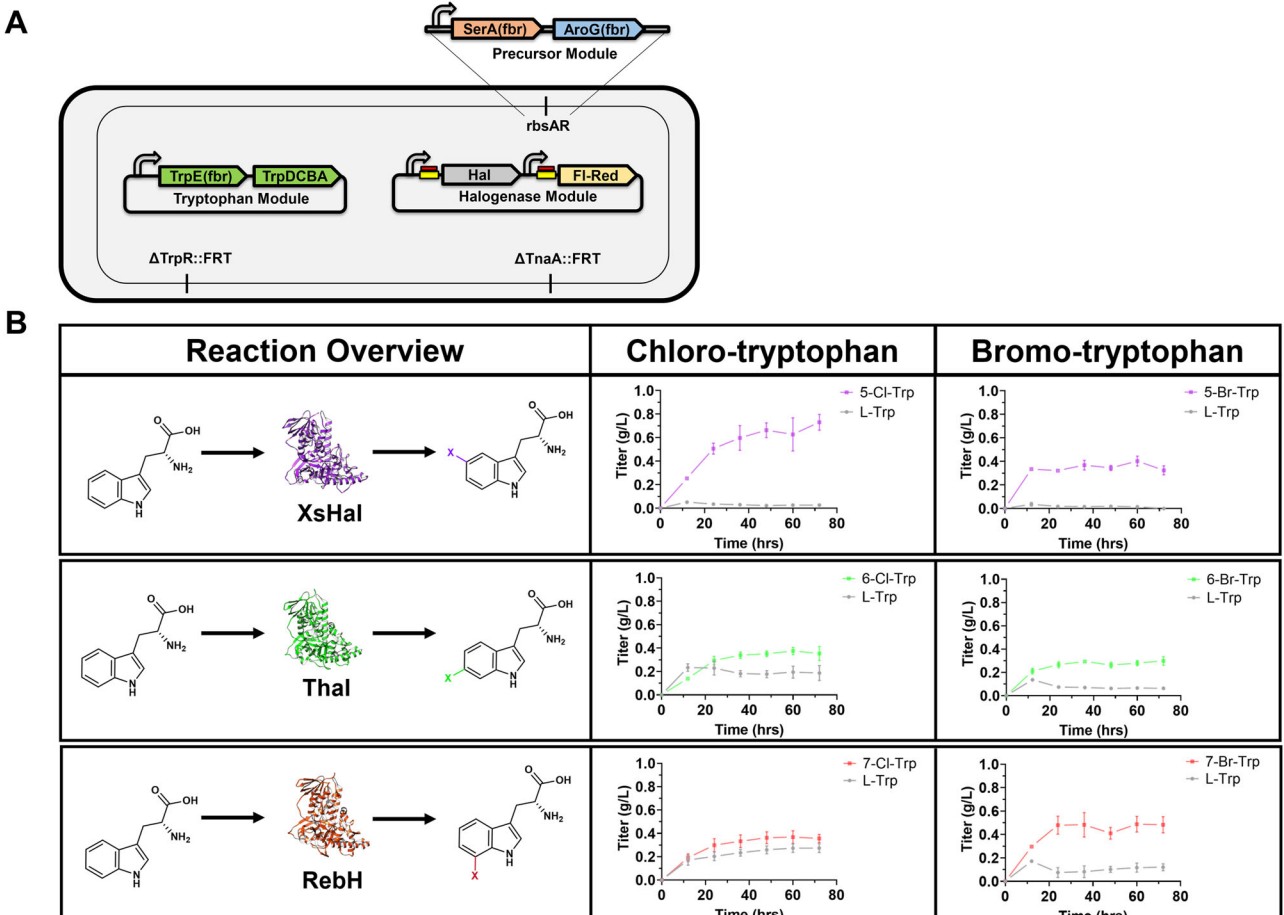

**Fig. 3 | De novo production of halogenated tryptophan in *E. coli*. A** Schematic of final halogenated tryptophan overproduction strain, showing modular engineering. Hal: halogenase (XsHal, ThaI, or RebH depending on the position of interest); Fl-Red: flavin reductase. **B** Reaction overviews of each introduced halogenase and corresponding production curves when fed with 40 g/L glucose for 72 h. Data are mean ± S.D.; $n = 3$ biological replicates. Source Data are provided as a Source Data file.

indole-3-acetamide, and characterized to be promiscuous as well, although unknown at the outset of this study[71]. TnaA, *E. coli*'s native tryptophan indole lyase, catalyzes the production of indole, a precursor for many other molecules including indigo and other important tryptophan dimers[72,73]. KynA catalyzes a ring-opening reaction to generate N-formyl-L-kynurenine and comprises the first step in the kynurenine and quinone pathways, classes of molecules with many bioactive characteristics[50,74,75]. McbB, one of the genes that drives the biosynthesis of marinacarbolines, was found in the organism *Marinactinospora thermotolerans* SCSIO 00652, and catalyzes a Pictet-Spengler cyclization process[76]. Beta-carbolines in general have been shown to have very interesting chemical characteristics including optoelectronic properties, potential as anti-cancer agents, and many other bioactivities[77–80]. Specifically, the molecules formed through the Mcb pathway in *Marinactinospora thermotolerans* have been shown to have antimalarial, cytotoxic, and anti-inflammatory activities[81,82]. TsrM catalyzes the reaction to 2-methyl-L-tryptophan via a unique cobalamin-dependent radical SAM mechanism and is the first step towards the synthesis of the antibiotic thiostrepton A[83–85]. Thus, this set showcases a wide range of biochemical reactions with fundamentally valuable end products.

Next, we sought to evaluate and harness the promiscuity of these downstream enzymes toward fed halogenated substrates in an effort to create diverse halogenated products. As outlined in Fig. 5A, the six selected downstream enzymes were each evaluated for their ability to convert six different halogenated tryptophan variants in the background of a *tnaA* deletion strain to prevent any unwanted product degradation. The binary promiscuity observations for these 42 combinations (when considering tryptophan and all 6 halogenated variants) are provided in Fig. 5B, whereas structures are displayed in Supplementary Fig. 47. From these assays, all 6 potential molecule sets **2b**, **2c**, **2d**, **2e**, **2f**, **2g** and **3b**, **3c**, **3d**, **3e**, **3f**, **3g** were observed when RgnTDC and iaaM were expressed, respectively. RgnTDC and iaaM are thus remarkably promiscuous enzymes, evidenced by their ability to convert all positions of halogenated tryptophan, supported by similar reports in prior literature[70,71]. All possible molecules **4b**, **4c**, **4d**, **4e**, **4f**, and **4g** were also observed when TnaA was expressed, thus echoing similar trends of promiscuity reports from this enzyme[53].

The remaining three enzymes were slightly less promiscuous, each making only 4 out of 6 possible halogenated downstream molecules. KynA was observed to readily convert 6-chloro (**1d**) and 6-bromo-tryptophan (**1e**) into 6-chloro-N-formyl-L-kyurenine (**5d**) and 6-bromo-N-formyl-L-kyurenine (**5e**), respectively, both precursors to prodrug 4-chloro- and 4-bromo-kynurenine[44,86] (Fig. 6A). KynA also exhibits slight promiscuity for the 5-Cl and 5-Br positions (molecules **5b** and **5c**) but molecules **5f** and **5g** were not observed. McbB has a much more complicated reaction mechanism and restrictive active site bundled between two subunits[87], yet was able to convert both the 5 and 7-position halo-tryptophan variants, molecules **1a**, **1b**, **1f**, and **1g**, to their corresponding carboline products, molecules **6b**, **6c**, **6f**, and **6g**, respectively with the 7-position produced at higher efficiency. Molecules **6d** and **6e** were not observed. Previous report have observed

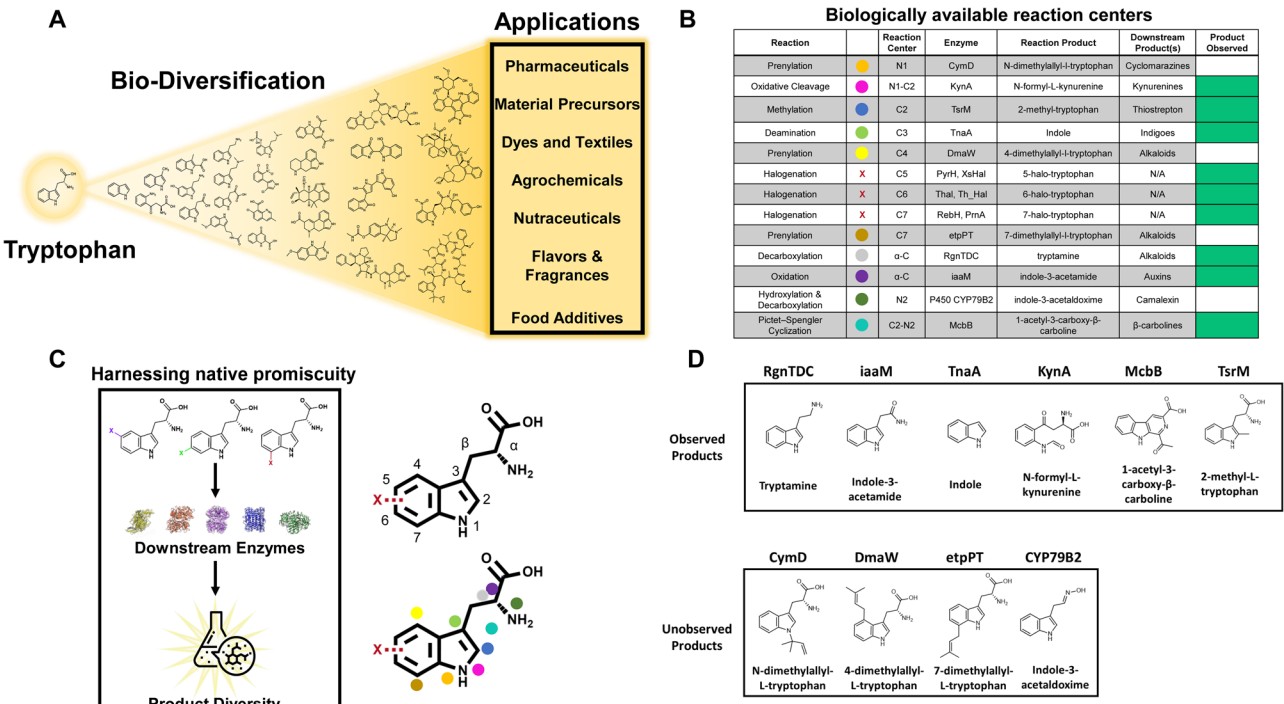

**Fig. 4 | Establishing downstream pathways to access molecules from tryptophan. A** Overview of example molecules derived from tryptophan and the applications enabled by them. Molecules in the figure (in general order left to right) include Indole, Kynurenine, Kynurenic acid, Skatole, 3-methyl-2-indolic acid (MIA), Serotonin, Indole-3-acetic acid (auxin), Tryptamine, indole pyruvic acid, 1-acetyl-3-carboxy-beta-carboline, strictosidine, indigo, violacein, rebeccamycin, thaxtomin, vinblastine, cyclomarin A, ergoline, ergotamine, N,N-Dimethyltryptamine, Hapalindole A, melatonin, psilocybine, lysergic acid, physostigmine, pyrrolnitrin, and quinmerac. **B** Biologically available reaction centers of tryptophan investigated within this study accessible through a single

enzymatic step. Note: This list is exemplary, not exhaustive. The modifications to the tryptophan scaffold are represented by colored reaction center dots, with colors corresponding to each reaction center displayed in (**C**). The green boxes in the product observed column represent the enzymes that converted tryptophan fed to the media into the expected product, as confirmed by LC-MS. **C** An overview of the halogen-product diversification strategy, utilizing promiscuity downstream enzymes to generate a wide range of halogenated tryptophan-derived products. **D** The chemical structure of the expected products are shown, with indications of observed and unobserved when 500 µM L-tryptophan was fed to the media.

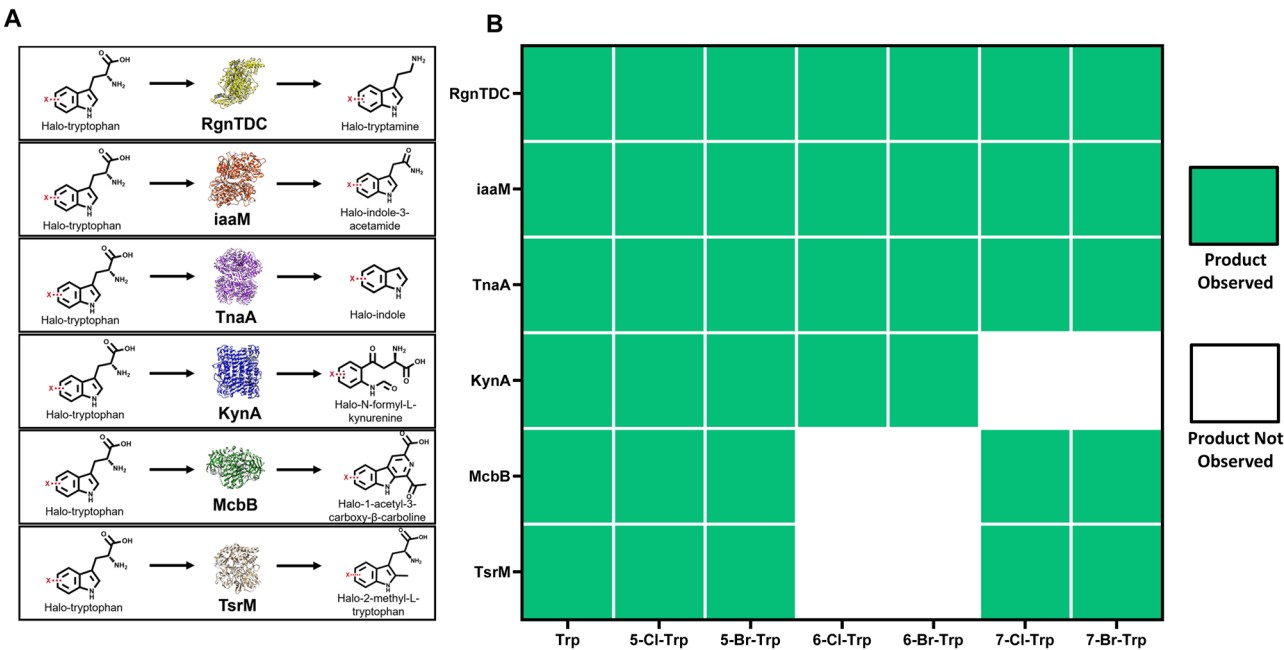

**Fig. 5 | Evaluation of downstream promiscuity through feeding assays.**
**A** Reaction overview for each downstream enzyme evaluated. **B** Each strain containing the corresponding downstream enzyme was fed 500 µM of each

halogenated tryptophan analog. Each heat map square represents the formation (green) or lack of formation (white) of corresponding halogenated products from biological triplicates picked from individual colonies.

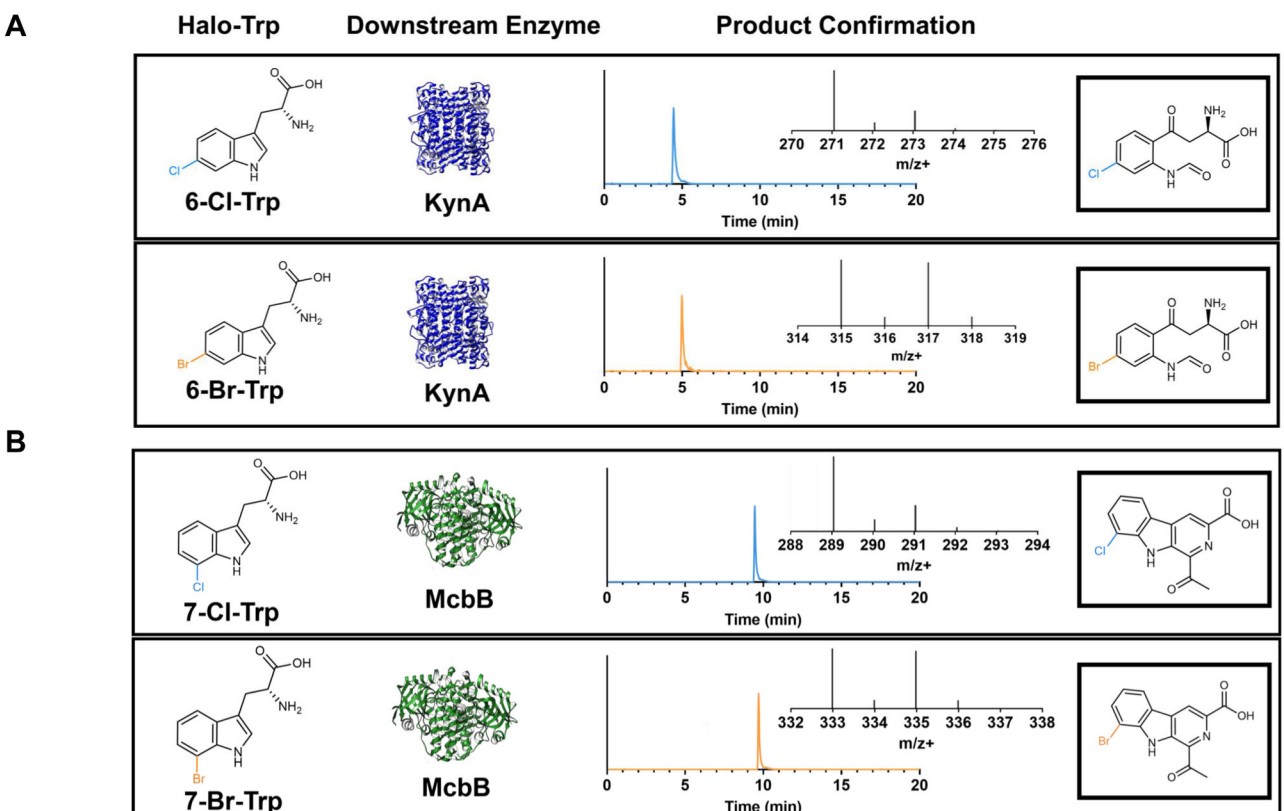

**Fig. 6 | Confirmation of pharmaceutically relevant molecules through microbial fermentation. A** Confirmation of prodrug precursors 6-chloro-N-formyl-L-kynurenine (blue) and 6-bromo-N-formyl-L-kynurenine (orange). **B** Confirmation of new-to-nature molecules 7-chloro-1-acetyl-3-carboxy-β-carboline (blue) and 7-bromo-1-acetyl-3-carboxy-β-carboline (orange).

McbB's ability to convert certain fluorine-modified tryptophans[76], though this represents confirmation of larger halogen substituted tryptophans. The four observed molecules are new-to-nature chloro- and bromo-modified beta-carbolines, with the 7-position highlighted in Fig. 6B. Lastly, TsrM follows a similar promiscuity pattern as McbB, where molecules **7b**, **7c**, **7f**, and **7g** were observed, corresponding to promiscuity for the 5 and 7-positions, whereas molecules **7d** and **7e** were not observed, reflecting similar trends seen in vitro[83]. The potential for di-halogenation of these downstream molecules was also investigated and was not evident based on the resulting LC-MS data. While we were not able to obtain analytical standards for the majority of downstream halogenated molecules, we have provided estimated titers and conversions of fed substrates from the feeding assay based on consumption of fed substrate (Supplementary Data 1 and 2).

These feeding assays coupled with downstream reaction promiscuity demonstrates the chemical diversity possible from tryptophan, whereby the tryptophan scaffold can be chemically decorated in a variety of ways. In general, we observed that promiscuity was related to the distance of the halogen from the specific reaction center in addition to the simplicity of the reaction mechanism. For example, RgnTDC and iaaM are remarkably promiscuous and both catalyze reactions near the α-carbon. TnaA is also highly promiscuous and catalyzes a simple cleavage reaction of the β-carbon bond from the indole side group. KynA, TsrM, and McbB on the other hand catalyze more complex reactions such as a complex ring opening, a methylation of the C-2 carbon, and ring closing reactions, respectively, within or very close to the indole side group. To further confirm our empirical observations, we pursued a computational investigation to determine relative binding energies between iaaM and McbB for tryptophan and halo-tryptophan precursors (Supplementary Discussion 2). General promiscuity trends were supported wherein iaaM has a larger binding pocket that can accommodate all variations of tryptophan and McbB's preference for 7-halo-tryptophan over the other positions is highly evident from the conformational change of a tyrosine group in the active site that especially occludes the 6-halo- position (Supplementary Table 1 and Supplementary Figs. 48–51).

## De novo production of halogenated products using synthetic, modular co-cultures

To enable true de novo production of rapidly diversified halogenated molecules, we sought to utilize a co-culture approach (Fig. 7), which represents a break from the convention of most previous downstream diversification of halogenated tryptophan works focused on combined pathway engineering within single cells[46,47,71]. Given that the halogenated tryptophan molecule is secreted from the cell, this point represents a natural break in metabolic pathways. Thus, combining a halogenated tryptophan overproduction strain with a downstream conversion strain can enable de novo production of an array of halogenated compounds. Moreover, we hypothesized that this co-culture approach was necessary as a consolidated bioprocessing approach (wherein halogenase and downstream enzyme are co-localized) would lead to high competition for the intracellular tryptophan pool by both competing pathways and result in more un-halogenated products. Similar approaches have been used to reduce metabolic burden and generate a variety of products downstream of tryptophan, such as tryptamine and indigo[88,89]. Thus, one cell can act to primarily produce halogenated tryptophan whereas the other cell can specialize on the downstream conversion of halogenated tryptophan.

To determine the effectiveness of the modular co-culture reactions, we compared the production of the non-halogenated downstream molecule using the wild-type tryptophan pool to that of the non-halogenated product formed during the co-culture reactions with

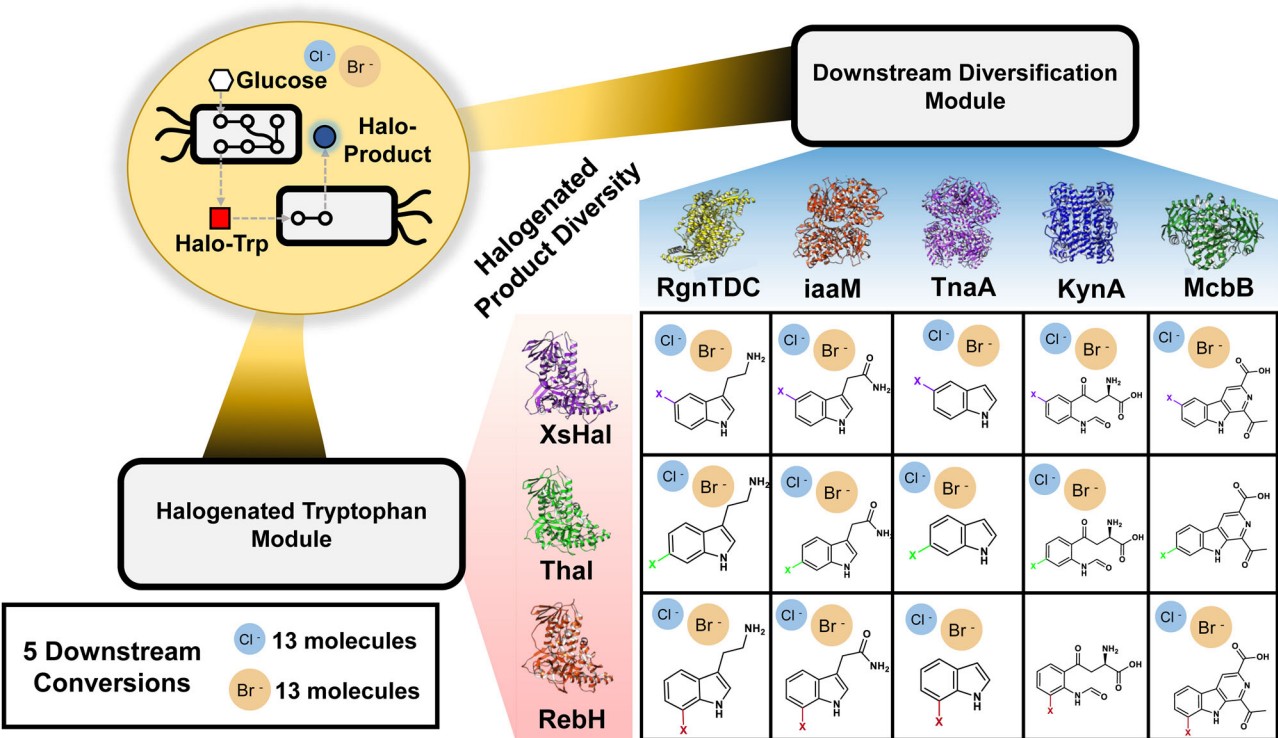

**Fig. 7 | Modular one-pot de novo co-culture reactions enable halogenated product diversity.** Schematic of the modular one-pot de novo co-culture reactions to enable halogenated product diversity. One strain converts glucose into 5-, 6-, or 7-halogenated tryptophan. The second strain converts the halogenated tryptophan into a halogenated downstream product which is secreted into the media. Boxes with both Br and Cl icons refer to downstream products for which both brominated and chlorinated versions have been detected using each respective downstream enzyme. Boxes without Br and Cl refer to downstream products which were not detected when attempting the specified co-culture.

both the halo-trp overproduction strain and downstream conversion strain. We evaluated this strategy for molecules where standards were available (5-chloro- and 5-bromo-tryptamine). As compared to a blank plasmid control, it was discovered that all the halo-tryptophan was readily converted into the equivalent halo-tryptamine molecule with a minimal increase in amount of the non-halogenated tryptamine product (Supplementary Figs. 52 and 53). Thus, the spatial separation effectively enables the generation of primarily the halogenated version. At the same time, we confirm de novo production of 36 mg/L of 5-chloro-tryptamine and 52 mg/L of 5-bromo-tryptamine at the 1 mL scale.

By deploying these modular, one-pot de novo co-culture reactions, we have confirmed the production of 26 distinct halogenated molecules from a glucose feedstock (Fig. 8). As expected, the de novo production results echo the feeding assays, where certain positions and enzymes are more promiscuous than others (Supplementary Figs. 54–84). TsrM was excluded from these experiments due to the low conversion of the native tryptophan substrate during the feeding assays. These de novo production schemes unlocked new access to synthesis from glucose in a microbial host of many of these products at the time the study was completed, highlighted with a circle (Fig. 8). Among these products include new-to-nature molecules, highlighted with a star, such as precursors to kynurenine prodrugs and the 5-chloro-, 5-bromo-, 7-chloro-, and 7-bromo-1-acetytl-3-carboxy-β-car-boline molecules that provide pathways to halogenated molecules that can serve as anti-inflammatory agents. While we were not able to obtain analytical standards for the majority of these products to enable full quantification, we have provided estimated titers of all products successfully produced via this co-culture format based on relative LCMS abundance areas and feeding assay estimated titers (Supplementary Data 2). These estimated titers have been benchmarked against exact titers for 5-Cl-Tryptamine and 5-Br-Tryptamine,

quantified via use of commercially available analytical standards (Supplementary Table 2). The estimated 5-Br-Tryptamine titer falls within 5% of the exact titer, while the estimated 5-Cl-Tryptamine titer exceeds the exact titer by ~50%. This indicates that the estimation accuracy likely varies with each compound. However, these estimates provide a general sense of titer scale (e.g., 1 mg/L vs 10 mg/L vs 100 mg/L), providing our best approximation in the absence of available reference standards for the majority of downstream halogenated compounds produced in this study. To provide clarity on the purity of compounds produced via co-culture, we have also provided estimated yields based on produced downstream molecule divided by total product, tryptophan and/or halo-tryptophan formed in the co-culture reaction (equal to product formed + residual tryptophan/halo-tryptophan; Supplementary Data 3).

## Discussion

Bioproduction offers a green solution for the generation of a diverse range of functional products. Herein, we have engineered *E. coli* as a microbial platform to convert glucose into a wide diversity of halogenated tryptophan derivatives. We showcase that through various engineering approaches, we can rapidly develop platform strains capable of producing six different halogenated tryptophan precursors de novo from glucose at high milligram-per-liter scales in flasks. This platform opens the door to many more applications and pushes the envelope towards reaching commercial viability for de novo biosynthesis of diverse halogenated molecules. By then investigating native promiscuity of six disparate downstream reactions in vivo, we show that many enzymes are amenable to further convert halogenated tryptophan precursors into a variety of halogenated downstream products in vivo, largely consistent with previous reports of promiscuity[48,50,53,70,71]. Lastly, we show that the halogenated tryptophan overproduction strains and strains equipped with downstream

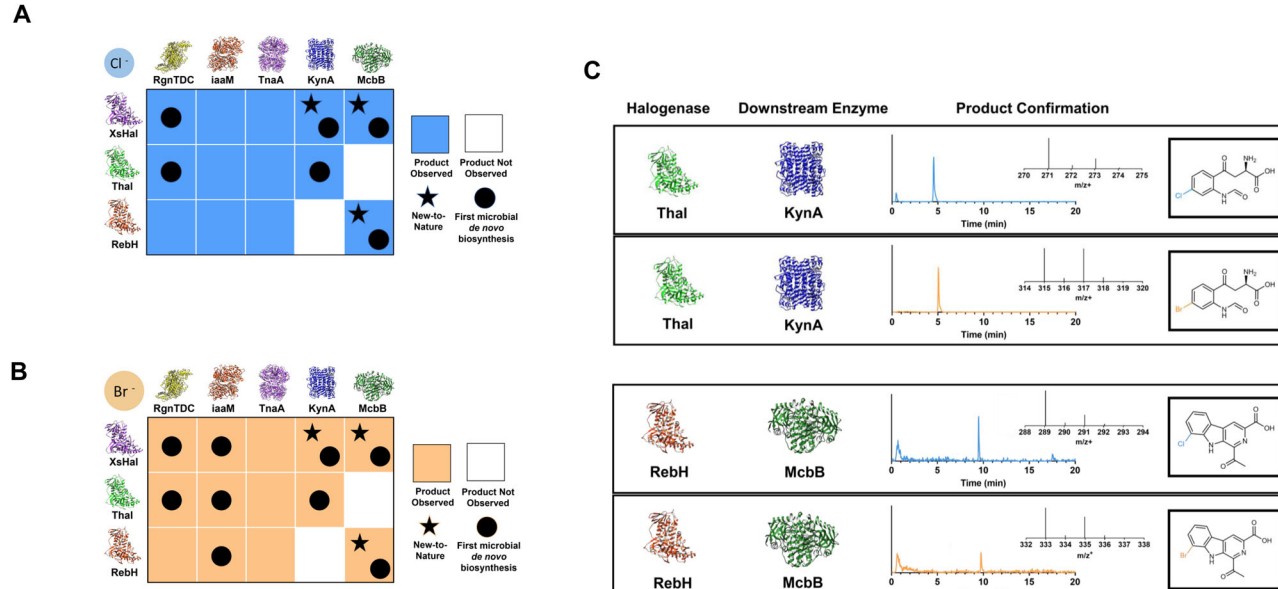

**Fig. 8 | De novo production of a wide range of tryptophan-derived halogenated products through modular one-pot co-cultures. A** Production of chloro-specific products, where each heat map square represents the formation (blue) or lack of formation (white) of corresponding halogenated products from biological triplicates picked from individual colonies. New-to-nature molecules are designated with a black star and products produced via de novo microbial synthesis for the first time are designated with a black circle. **B** production of bromo-specific products, where each heat map square represents the formation (orange) or lack of formation (white) of corresponding halogenated products from biological triplicates picked from individual colonies. New-to-nature molecules are designated with a black star and products produced by microbial synthesis are designated with a black circle. **C** Confirmation of de novo production of 6-chloro-N-formyl-L-kynurenine (5d), 6-bromo-N-formyl-L-kynurenine (5e), 7-chloro-1-acetyl-3-carboxy-β-carboline (6f), 7-bromo-1-acetyl-3-carboxy-β-carboline (6g), respectively, via LC-MS.

enzymes can be combined in a modular co-culture fashion to generate over 26 distinct halogenated molecules including 15 first-time de novo biosynthesized, of which 6 are entirely new-to-nature products. Future investigation of promiscuity for the pathway enzymes is warranted, where quantification via in vitro reactions could enable a better understanding of each enzyme's ability to turnover halogenated tryptophan variants. Additionally, pursuits to engineer and evolve native selectivity in favor of the halogenated variant of interest could prove fruitful, similar to the enzymes that have already evolved in nature, such as Tar13, that are more specific for halogenated tryptophan than for tryptophan[44]. Taken together, this platform demonstrates a synthetic bio-combinatorial chemistry approach to yield de novo production of halogenated compounds of relevance to a variety of industrial sectors.

## Methods

### Chemicals and materials

L-tryptophan, indole, 5-chloroindole, 6-chloroindole, 7-chloroindole, 5-bromoindole, 6-bromoindole and 7-bromoindole were purchased from Sigma-Aldrich. 5-chloro-L-tryptophan, 5-bromo-L-tryptophan, 6-chloro-L-tryptophan, 7-chloro-L-tryptophan, and 7-bromo-L-tryptophan were purchased from Advanced Chemblocks, Inc. 6-bromo-L-tryptophan was purchased from Santa Cruz Biotechnology. D/L versions were purchased from the same vendor. M9, Minimal Salts, 5X was purchased through Sigma-Aldrich.

### Strains, plasmids, and transformations

*Escherichia coli* strain BW25113 with deletions of *tnaA* or *trpR* were obtained from *E. coli* Genetic Resources at Yale CGSC, The Coli Genetic Stock Center. Bacterial genomic DNA was extracted using the Wizard Genomic DNA Purification Kit (Promega). Lamba Red Recombination was used for integration of relevant cassettes (Upstream Module) which has been well-documented elsewhere[90]. pHal, with pBR322 origin and TacI promoter was used for all final experiments. Plasmids were assembled using either Gibson's method, ligation, or a modified

Golden Gate cloning method to produce plasmids[91]. Plasmids, primers, ORFs, and other genetic elements are listed in Supplementary Tables 3 and 4 as well as Supplementary Data 4. Various PCR products were then inserted into this plasmid under control of the strong Ptac promoter for overexpression in *E. coli*. These plasmids are referred to as pHal throughout this work. The inserts were PCR amplified using Q5 Hot Start High-Fidelity DNA Polymerase (NEB). Cells were then electroporated and recovered in 1 mL SOC at 37 °C for 1 h. A small portion was plated on LB+Amp+Chl plates to check for transformation efficiency, while the rest was moved to LB+Amp+Chl to grow overnight. Cultures were then freezer stocked and added to screening plates the next day.

### Halogenase and flavin reductase expression

Strains with specified plasmids were typically grown in LB with appropriate antibiotics overnight. The next day, cultures were diluted back to an OD of 0.1 and allowed to grow for 2 h at 37 °C until OD reached 0.7–0.9. 1 mM IPTG was then added and cultures were grown at 30 °C for 2 h to induce expression of halogenase and flavin reductase. Cultures were spun down and media was replaced with M9 salts, 5 g/L glucose, and 1 g/L casamino acids with appropriate antibiotics, 1 mM IPTG, and 1 mM L-tryptophan. Suspension cultures were grown in Fisherbrand™ 96-Well DeepWell™ Polypropylene Microplates and incubated using an Infors HT Multitron Pro with 1000 rpm shaking.

### Halogenase panel experimental conditions

Strains containing pHal-ThFre-Hal (various halogenases) were grown in LB with Amp (100 μg/mL) overnight. The next day, cultures were diluted back to an OD of 0.1 and allowed to grow for 2 h at 37 °C until OD reached 0.7–0.9. 1 mM IPTG was then added and cultures were grown at 30 °C for 2 h to induce expression of halogenase and flavin reductase. Cultures were spun down and media was replaced with M9G + CAA with appropriate antibiotics, 1 mM IPTG, and 1 mM L-tryptophan. Suspension cultures were grown in Fisherbrand™ 96-

Well DeepWell™ Polypropylene Microplates and incubated using an Infors HT Multitron Pro with 1000 rpm shaking.

### De novo production of halogenated tryptophan in *E. coli*

Strains containing specified modifications were grown up in LB with any necessary antibiotics overnight at 30 °C. The next day, these strains were diluted back 100-fold to OD ~ 0.1 and allowed to grow to OD 0.7–0.9 in 25 mL LB media at 37 °C in a 250 mL shake flask. Cultures were then induced with 1 mM IPTG and allowed to grow for 2 h at 30 °C. Cultures were then spun down at 3000 × *g* for 10 min in Falcon tubes to remove LB media and were replaced with 25 mL of M9G media (M9 salts, 40 g/L glucose) and placed in a 250 mL shake flask. These were then allowed to grow up for 72 h with timepoints taken every 12 h to be analyzed on HPLC.

### Downstream promiscuity feeding assays

Strains containing corresponding downstream enzymes were grown up in LB+Chl (34 µg/mL) overnight. Strains were then inoculated at OD ~ 0.1 in M9G + CAA+Chl and allowed to grow for 2 h in which 1 mM IPTG and 500 µM of corresponding halogenated tryptophan analogs were added to the media. Downstream products were then confirmed via LC-MS.

### Downstream promiscuity docking studies

The Rosetta software suite is a platform for the computational modeling of protein structures. PyRosetta, a Python binding for Rosetta[92], was utilized to compare the structures of iaaM and McbB (Protein Data Bank accession codes of 4iv9 and 3 × 27, respectively), as these crystal structures are the only structures in this study to contain bound tryptophan. An ensemble of conformations in ".mol" format were generated for 5-chloro-, 5-bromo-, 6-chloro-, 6-bromo-, 7-chloro-, and 7-bromo-tryptophan using the OpenBabel chemical toolbox. These were then converted into a ".param" file and several ".pdb" files for use with PyRosetta. Each halogenated tryptophan was aligned to the native tryptophan binding mode in each enzyme crystal structure, resulting in 14 complexes including the original complex. Each complex was minimized into Rosetta's energy scoring system using the "ref2015_cart" scorefunction and FastRelax. Finally, each complex was analyzed using the InterfaceAnalyzerMover to determine an overall binding energy or ΔΔG. All code used to generate the structures and binding scores can be found at https://github.com/jordantwells42/downstream-docking.

### De novo co-culture production of halogenated tryptophan derivatives in E. coli

Strains containing specified modifications were grown up in LB with any necessary antibiotics overnight at 30 °C. The next day, these strains were diluted back 100-fold to OD ~ 0.1 and allowed to grow to OD 0.7–0.9 in 1 mL LB media at 37 °C. Cultures were then induced with 1 mM IPTG and allowed to grow for 2 h at 30 °C. Cultures were then spun down at 3000 × *g* for 10 min to remove LB media and were replaced with 500 µL of M9G media with micronutrients (M9 salts, 40 g/L glucose, 1X micronutrient solution), then combined together in a single reaction of 1 mL total volume. These were then allowed to incubate for 48 h, where the supernatants were spun down and analyzed on HPLC.

### Quantification of products

Cultures were typically grown at 30 °C in a shaking incubator at 1000 rpm for specified amounts of time (0-48 h). After specified amounts of time, OD600 was measured using a Tecan plate reader as necessary. Cultures were then centrifuged at 3000 × *g* for 10 min to pellet the cells, and the supernatant was removed for further analysis. Metabolite quantification was performed on HPLC or LC-MS using authentic standards, depending on availability. Supernatants were

then submitted for LC-MS analysis to confirm the presence of the expected downstream products for each reaction or compared to authentic standards on HPLC for the case of indole analogs, which had difficulty fragmenting on the MS, even at high concentrations of an authentic standard. Quantification was performed on a Dionex Ulti-Mate 3000 (Thermo) equipped with an LS Eclipse Plus C18 column (3.0 × 150 mm, 3.5 µm; Agilent). The mobile phase for tryptophan and halogenated tryptophan analysis consisted of 1% (v/v) acetic acid in water or acetonitrile. Detection was performed at 280 nm with a flow rate of 0.3 mL min⁻¹ and a column temperature of 30 °C. Data processing was performed using Chromeleon software. Calibration standards were prepared for tryptophan and halogenated tryptophan. Downstream molecules were detected on an LC-MS. Sample supernatants were loaded directed into the instrument without additional preparation. All measurements were performed in biological triplicate with representative spectra displayed in the supplemental information. For LC/MS analysis, the samples were analyzed using an Agilent 6546 A Q-TOF interfaced with an Agilent 1260 Infinity II liquid chromatography system (G7112B) and an Agilent Dual Jet Stream electrospray ionization (ESI) source (G1958-65271). The mass spectrometry conditions were as follows: autosampler temperature 7 °C; column temperature 30 °C; electrospray ionization in positive mode; capillary voltage 3500 V; nozzle voltage 2000 V; fragmentor voltage 80 V; nitrogen drying gas temperature 350 °C; nitrogen drying gas flow rate 10 L/min; sheath gas temperature 350 °C; sheath gas flow rate 11 L/min; nebulizer pressure 60 psi; mass range 50–1000 *m/z*. LC separations were achieved on an Agilent Rapid Resolution HD ZORBAX Eclipse Plus C18 column (P.N. 959757-902: 50 × 2.1 mm, 1.8 micron particle size) preceded by an Agilent ZORBAX Eclipse Plus C18 narrow bore guard column (P.N. 821125-936: 12.5 × 2.1 mm, 5 micron particle size). The LC conditions were as follows: solvent A was Water with 0.1% formic acid; solvent B was Acetonitrile; flow rate 0.4 mL/min; gradient ramp held 5% B for 2 min, ramped to 20% B from 2 to 5 min, ramped to 95% B from 5 to 12 min, held at 95% B until 16 min, then re-equilibrated at 5% B from 16.1 to 20 min. LC/MS data were collected using Agilent MassHunter Workstation LC/MS Data Acquisition for 6500 series Q-TOF (Version 10.1) and analyzed using Agilent MassHunter Workstation Qualitative Analysis (Version 10.0) software. All *m/z* values and spectra were calculated and collected based on the expected structures of the respective compounds of interest using MassHunter's internal search function.

### Reporting summary

Further information on research design is available in the Nature Portfolio Reporting Summary linked to this article.

## Data availability

Data supporting the findings of this work are available within the paper and its Supplementary Information file. A reporting summary for this Article is available as a Supplementary Information file. Source data are provided with this paper.

## Code availability

All code used in this study can be found at Github [https://github.com/jordantwells42/downstream-docking].

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

## Acknowledgements

This work developed in part with funding through a subcontract from Zymergen under the Defense Advanced Research Projects Agency's (DARPA) Living Foundries: 1000 Molecule program. The views, opinions, and/or findings expressed are those of the author and should not be interpreted as representing the official views or policies of the Department of Defense or the U.S. Government. Sequencing was conducted at the Genomic Sequencing and Analysis Facility (RRID no. SCR_021713). LC-MS and other analytical characterizations were conducted at the UT Mass Spectrometry Facility. K.B.R. and S.M.B. acknowledge a National Science Foundation (NSF) Graduate Fellowship.

## Author contributions

H.S.A. and K.B.R. conceived the project idea. K.B.R., H.S.A., and S.M.B. designed and directed the research. K.B.R. and H.S.A. wrote the manuscript. K.B.R., H.S.A., and S.M.B. revised the manuscript. K.B.R. and S.M.B. carried out experiments and interpreted the results under the guidance and direction of H.S.A. J.W. performed docking simulations. K.J.B. performed LC/MS analysis and compiled results. M.Z., K.P., S.D., A.T., and S.G. contributed to gene construction, culturing, and analysis. All authors reviewed the final version of the manuscript.

## Competing interests

The authors declare no competing interests.
