## [Peer Review File · Nature Communications]

A Modular and Synthetic Biosynthesis Platform for de novo Production of Diverse Halogenated Tryptophan-Derived MoleculesEditorial Note: This manuscript has been previously reviewed at another journal that is not operating a transparent peer review scheme. This document only contains reviewer comments and rebuttal letters for versions considered at Nature Communications.

Reviewers' Comments:

Reviewer #3:

Remarks to the Author:

Due to the unavailability of the prior reviewers for the re-assessment of the manuscript by Reed et al. I was asked on my opinion whether the authors have appropriately addressed the remaining comments from the previous reviewers and on my opinion on the overall suitability of the manuscript for publication in Nature Communications.

My overall assessment of the manuscript can be found directly below. My opinion on the completeness of the author's responses to the comments of the previous reviewers can be found highlighted in red after each comment in the point-by-point reply (attached document).

Overall assessment:

Reed et al present an interesting manuscript which describes the de novo synthesis of a range of halogenated tryptophans in *Escherichia coli* from glucose as a starting material. In this work, the authors have leveraged both well-known (e.g., RebH and Thal) and a new flavin-dependent halogenase (XsHal) by incorporating them into an engineered strain of *E. coli* capable of producing 300-700 mg/L of halogenated tryptophans. They subsequently identified a set of downstream enzymes that can act on the halogenated tryptophans and report the biosynthesis of 26 distinct halogenated compounds.

Having reviewed the authors point-by-point reply to the reviewer's comments, I feel that they have only partially addressed the concerns of the reviewers. While I think the authors have taken reviewer 1's concerns regarding referencing and context into consideration by adapting the main text, they have not incorporated a measure of conversion or yield for the downstream enzymatic products reported. Furthermore, the products have not been characterized or compared to authentic reference standards. This makes it very difficult to benchmark their system.

I would also agree with the overall impression of the previous reviewers, that - given the wealth of literature which is available on the enzymatic halogenation of tryptophans, in addition to their ability to be further modified by downstream enzymes - this manuscript lacks the novelty required for publication in Nature Communications.

Additional comments to figures and supplementary figures:

- Line 189: Table 2 cited in the text is not provided
- Line 225: It would be useful to reiterate the figure no. when referring to "shown in white in the Product Observed column"
- Supplementary figures 5 and 6: It would be useful to label residues shown in docking studies - especially the Tyr216 which undergoes a conformational change in McbB.

Response to Reviewer Comments

****Note:** Responses to reviewer comments in this round of review are listed in **Purple****

Reviewer #3 (Remarks to the Author):

Due to the unavailability of the prior reviewers for the re-assessment of the manuscript by Reed et al. I was asked on my opinion whether the authors have appropriately addressed the remaining comments from the previous reviewers and on my opinion on the overall suitability of the manuscript for publication in Nature Communications.

My overall assessment of the manuscript can be found directly below. My opinion on the completeness of the author's responses to the comments of the previous reviewers can be found highlighted in red after each comment in the point-by-point reply (attached document).

Overall assessment:

Reed et al present an interesting manuscript which describes the de novo synthesis of a range of halogenated tryptophans in Escherichia coli from glucose as a starting material. In this work, the authors have leveraged both well-known (e.g., RebH and Thal) and a new flavin-dependent halogenase (XsHal) by incorporating them into an engineered strain of E.coli capable of producing 300-700 mg/L of halogenated tryptophans. They subsequently identified a set of downstream enzymes that can act on the halogenated tryptophans and report the biosynthesis of 26 distinct halogenated compounds.

Having reviewed the authors point-by-point reply to the reviewer's comments, I feel that they have only partially addressed the concerns of the reviewers. While I think the authors have taken reviewer 1's concerns regarding referencing and context into consideration by adapting the main text, they have not incorporated a measure of conversion or yield for the downstream enzymatic products reported. Furthermore, the products have not been characterized or compared to authentic reference standards. This makes it very difficult to benchmark their system.

I would also agree with the overall impression of the previous reviewers, that - given the wealth of literature which is available on the enzymatic halogenation of tryptophans, in addition to their ability to be further modified by downstream enzymes – this manuscript lacks the novelty required for publication in Nature Communications.

Response: We thank the reviewer for this overview and review of the manuscript. To further address the prior reviewers' comments, we have incorporated measurements of conversion and yield for appropriate experiments into the supplementary information and provide some additional details listed in the response to the specific point-by-point comments below. It should be noted that we are unfortunately unable to access authentic reference standards for the vast majority of downstream halogenated tryptophan products made *de novo* in this work and thus cannot provide exact titers. As a result, we have provided estimates for titers, conversions, and yields using assumptions based on amount of fed precursor consumed, as well as comparisons of LCMS abundance for molecules in the feeding assays and coculture experiments. Full details for these assumptions and calculations are listed below in this document (in the updated response to Reviewer 1's Comment 8) as well as in the revised manuscript. As previously discussed, while prior works have focused on specific tryptophan halogenation reactions or downstream modifications, we believe this to be the first and most comprehensive work fully showcasing the combination of metabolic engineering, coculture

design, and combinatorial chemistry to access a wide array (26 distinct downstream products) of halo-tryptophan-derived compounds in a purely *de novo* fashion from simple sugar starting materials. We thus feel the study is novel compared to prior works.

Additional comments to figures and supplementary figures:

- Line 189: Table 2 cited in the text is not provided

Response: We thank the reviewer for this comment. Table 2 has been provided in the revised version of the manuscript.

- Line 225: It would be useful to reiterate the figure no. when referring to “shown in white in the Product Observed column”

Response: We thank the reviewer for this comment. We have updated the text to include a reference to Figure 3b.

- Supplementary figures 5 and 6: It would be useful to label residues shown in docking studies – especially the Tyr216 which undergoes a conformational change in McbB.

Response: We thank the reviewer for this comment. We have labeled the Tyr216 in Supplemental Figures 5 and 6, as well as denoted whether it is in the up or down conformation. This should add clarity to the reader.

Reviewer 3 analysis of prior responses to Reviewer 1 and Reviewer 2:

Reviewer 1:

Dear Colleagues,

I am writing to provide a review of Nat. Chem. Biol. manuscript NCHEMB-A230316286. If your review policy permits, please provide my signed review in full to the authors; if your policy precludes this possibility, I encourage you to reconsider that policy. In my experience, taking ownership of a review makes it more thoughtful, more objective, and more collegial.

This study describes how tryptophan halogenases can be introduced into strain of E. coli engineered for tryptophan production to produce halogenated tryptophans from glucose in titers around 0.4-0.8 g/L (Fig. 3). It then shows how co-culturing these strains with others containing enzymes that can act on halogenated tryptophan can be used to generate halogenated derivatives of different compounds. Flavin-dependent halogenase biocatalysis, including using these enzymes for metabolic engineering, is a well-established field. As outlined below, the central goal of this study, "de novo production" of halogenated tryptophan and derivatives generated via downstream enzymes, has been achieved by multiple groups though this precedent was either not clearly explained or omitted. The key advance that the authors made was to use a strain of E. coli containing mutations known to increase tryptophan production (refs 54-59) and to then optimize this strain. To my knowledge the co-culturing approach is also new to halogenase biocatalysis. I regard these as technical advances that, while useful, do not represent a significant break from fairly extensive precedent discussed below. I also believe that the authors need to provide better characterization of the products formed because the data provided do not give any sense of product yield or purity, which mitigates the potential utility of their approach.

1. The authors state that "The combinatorial biochemistry afforded by linking halogenases with downstream enzymes can access a diverse array of halogenated compounds." Note, reference 37 isn't relevant as it deals with conventional synthetic chemistry methods and should be omitted. After listing some examples, the authors claim that "While these examples demonstrate advances in halogenated metabolism, they do not ubiquitously describe *de novo* production, an important consideration given the relatively expensive and low-water-soluble substrates used in these studies such as indole, tryptophan, or other halogenated precursor molecules." This passage is misleading and omits key precedent. For example, references 40 and 45 do not demonstrate "*de novo* production" in part because the authors were using substrates that cannot be accessed via biocatalysis, but they do show *in vivo* conversion of these derivatives to other products, so this distinction needs to be made. References 41-44, however, do involve *de novo* synthesis unless I am misunderstanding them. Reference 41 describes "*de novo* production" of halogenated tryptophan and alkaloids in *C. roseus*: "If prokaryotic halogenases could function in the eukaryotic plant cell, and if tryptophan decarboxylase could convert halogenated tryptophan into halogenated tryptamine, then *C. roseus* would produce chlorinated alkaloids *de novo*" (both "ifs" were validated). Reference 42 notes "To attain the *de novo* biosynthesis in a plant-based system devoid of indican, we employed a sequence of enzymes from diverse sources, including three microbial tryptophan halogenases substituting the amino acid at either C5, C6, or C7 of the indole moiety." Reference 43 describes both *de novo* synthesis of halogenated Trp derivatives in planta and the use of synthetic Trp substrates to access non-natural derivatives as noted above: "Exogenous tryptophan substrates, if used [emphasis added], were infiltrated by the same method on the 4th day of growth as 100 μ M solutions in infiltration buffer". Reference 44 involves *de novo* production of various Trp derived compounds, including halogenated kynurenines and quinolines in yeast. The cells were grown in media containing peptone, so if the authors are discounting this example because of the media used, they should make that clear. The authors do not cite an early example of combinatorial *de novo* biosynthesis of halogenated indolocarbazoles in *S. albus* (DOI: 10.1073/pnas.0407809102), so this should be done. These authors indicate that R5 media was used and that some experiments were fed with different indolocarbazole precursors, but the latter are not relevant to the halogenated products reported since they are not substrates for FDHs. Goss has also reported various efforts aimed at generating halogenated tryptophan derivatives in *E. coli* for subsequent transition metal catalysis (DOI: 10.1038/s41467-017-00194-3): "we determined that a minimal media containing potassium nitrate (20 mM) as the nitrogen source, glycerol (0.5% w/v, 55 mM) as the sole carbon source and supplemented with 50 mM NaBr ('cross-coupling media' (CCM)) could be employed." Note, this study also reports studies supplementing media with Trp, but I confirmed with Goss via email that they do not need to add Trp. This extensive precedent for *de novo* synthesis of halogenated Trp isomers and other compounds via *in vivo* enzymatic elaboration needs to be clarified. These studies also used two of the three FDHs used for the bulk of the studies in the current manuscript. In my opinion, this precedent significantly weakens the novelty of the current study.

We thank the reviewer for the overview and comment. We have taken these comments regarding references and background/context into consideration and improved upon the manuscript. The premise of our study is a demonstration of the power of leveraging combined efforts in metabolic engineering, coculture design, and combinatorial chemistry to access a wide array of halo-tryptophan-derived compounds (26 distinct final downstream products in total) via whole cell biocatalysis from glucose as a starting material. We feel the study to be sufficiently comprehensive and novel as compared to prior works. These additions outlined here help to contextualize the

work better.

Regarding the reference 37, this is referenced in a sentence that states “ Tryptophan itself serves as a gateway to a plethora of interesting natural products ranging from small molecules like indole, kynurenines, quinones, and tryptamines to larger molecules like violacein, strictosidine, and beta-carbolines 36–39”. In doing so, this sentence is referring to the breadth of products which can be derived from tryptophan as a substrate, not limited to biochemistry vs synthetic chemistry. Thus, we feel it is appropriate to keep ref 37 there to demonstrate the importance of tryptophan as an important precursor.

Regarding the comments on precedent for de novo production, we have edited the statement in the text to say “While these examples demonstrate advances in halogenated metabolism, they do not ubiquitously describe de novo microbial production of diverse halogenated tryptophan- derived compounds starting from simple sugar starting materials. This is an important consideration given the relatively expensive and low-water-soluble substrates often used in prior studies such as indole, tryptophan, or other halogenated precursor molecules, as well as need for de novo production in planta to rely on seasonally-dependent crop yields”. This edit points to the fact that our main emphasis is on production in microbial hosts from simple sugar starting materials (i.e. glucose), which has not been widely demonstrated for the full breadth of products of interest in this study.

Finally, we have added the suggested studies on production of halogenated indolocarbazoles, as well as production of halogenated tryptophan derivatives for subsequent transition metal catalysis, to the main text.

The reviewer’s comment has been adequately addressed in the point-by-point reply and addressed within the manuscript.

Response: We thank the reviewer for this feedback.

*2. The authors note that "The first gram-scale production of 7-chloro-tryptophan from a tryptophan feed was demonstrated in vitro using cross-linked enzyme aggregates (CLEA's)³⁴. Recent efforts in the engineering of *Corynebacterium glutamicum* have demonstrated the first de novo and in vivo gram- scale production of 7-bromo-tryptophan. These studies provide promise for expanding bio-based production to alternative hosts and halogenation positions within tryptophan." It is somewhat misleading to cite this one example involving "de novo and in vivo gram-scale" Trp halogenation given that there are many additional examples of de novo and in vivo Trp halogenation, as discussed above. The authors should clarify this distinction.*

We thank the reviewer for this comment. We agree that there are other examples of tryptophan halogenation, especially coupled with additional downstream modifications, which we cite in the following paragraph in the text in which we are discussing combinatorial biochemistry. The rationale for citing these studies which showcase accounts of ‘gram-scale’ production of halogenated tryptophan at this point in the text is to call out the potential for production at industrially relevant scale. To avoid confusion of this point we have altered the text to say “These studies provide promise for expanding bio-based production to alternative hosts and halogenation

positions within tryptophan at a scale relevant for commercially viable industrial production.

The reviewer's comment has been adequately addressed in the point-by-point reply and addressed within the manuscript.

Response: We thank the reviewer for this feedback.

3. The authors claim that "While de novo production of halogenated tryptophan has been reported in C. glutamicum for 7-Br-tryptophan and detectable quantities of 7-Cltryptophan^{35,46}, no studies have reported such production (for these particular halogenated forms or others) in E. coli." As noted above, Goss has done this. The authors should conduct a careful literature search to ensure that they aren't missing other examples.

We thank the reviewer for this comment. The study by Goss that is referred to lists 1.5 mM L-Tryptophan as a media component to then produce 7-Br-Trp. While it is great that the reviewer has confirmed via email that the tryptophan supplementation is not necessary, we cannot cite email communication as peer-reviewed citations in the text and thus must go with what is reported in the paper. To our knowledge, no other work has reported appreciable (i.e. close to gram-scale titers) amounts of halogenated tryptophan produced de novo in E. coli. To avoid confusion we have edited the text to emphasize this lack of reported de novo, high-titer production: "While de novo production of halogenated tryptophan has been reported in C. glutamicum for 7-Br-tryptophan and detectable quantities of 7-Cl-tryptophan^{35,48}, no studies have reported such de novo production (for these particular halogenated forms or others) of close to gram-scale titers in E. coli.". We have conducted a review of the literature in this regard.

The reviewer's comment has been adequately addressed in the point-by-point reply however the sentence outlined above has not been incorporated into the main text.

Response: We thank the reviewer for this comment and for catching this mistake. This sentence has been added to the revised version.

4. The authors state that "Expression of a heterologous, more thermostable flavin reductase (Th-Fre) resulted in higher production over EcFre and the null strain, showcasing the importance of optimizing the cofactor balance for the halogenase reaction (Supplementary Fig. 1)." Several groups have already demonstrated the importance of cofactor supply for in vivo halogenation. The Goss and O'Connor studies noted above do this, and my group reported that fusing a flavin reductase to a halogenase improves yields of halogenated Trp in vivo (DOI:10.1002/cbic.201700391). These studies (and other relevant ones) need to be acknowledged and the sentence should be edited so that it indicates that the current finding is consistent with previous studies rather than "showcasing" a new finding.

We thank the reviewer for this comment. We have cited the suggested studies, plus others we have found via a literature search, to highlight prior work demonstrating the importance of cofactor balancing for tryptophan halogenase reactions. We have edited the sentence in the text to state "Expression of a heterologous, more thermostable flavin reductase (Th-Fre) resulted in higher production over EcFre and the null strain, reinforcing the importance of optimizing the cofactor balance for the halogenase reaction as noted in prior studies 34,42,43,50,51" to avoid confusion

in this regard.

The reviewer's comment has been adequately addressed in the point-by-point reply however the sentence outlined above has not been incorporated into the main text.

Response: We thank the reviewer for this comment and for catching this error. The full sentence is now included and updated with the current document references. It reads “Expression of a heterologous, more thermostable flavin reductase (Th-Fre) resulted in higher production over EcFre and the null strain, reinforcing the importance of optimizing the cofactor balance for the halogenase reaction as noted in prior studies^{36–38,46,47}.”

5. *The authors state that "the halogenase XsHal, which has not been previously expressed in vivo in a heterologous host, displayed robust production of both 5-chloro- and 5-bromo-tryptophan precursors." Reference 53 describes heterologous expression of this enzyme in E. coli. If the authors mean that XsHal has not been used for in vivo halogenation, that should be clarified.*

We thank the reviewer for this comment. Yes, we were referring specifically to use for in vivo production as opposed to expression in E. coli followed by extraction and in vitro use. We have clarified the text to state ‘the halogenase XsHal, which has not been previously used for in vivo production using a heterologous host, displayed robust production of both 5-chloro- and 5-bromo-tryptophan precursors’ to avoid confusion in this regard.

The reviewer's comment has been adequately addressed in the point-by-point reply and addressed within the manuscript.

Response: We thank the reviewer for this feedback.

6. *The authors state that "These strains achieve a selectivity reaching as high as 96% for bromotryptophan even in the presence of competition for the halide salt with the residual chloride present in the growth media (Supplementary Fig. 3)." Similar selectivity has been reported in some of the studies noted above, so again, the authors should clarify that their finding is consistent with previous reports.*

We thank the reviewer for this comment. Many previous works do not directly report selectivity and without sufficient media optimization during the fermentation, the selectivity can be much lower (Supplementary Figure 3). We believe that this is a useful metric to report and acknowledge that similar selectivity has been found in other studies with other organisms including C. glutamicum so we have clarified the text to state “...(Supplementary Fig. 3), consistent with previous reports in other organisms that use similar levels of bromide salt in the media 35

The reviewer's comment has been adequately addressed in the point-by-point reply.

Response: We thank the reviewer for this feedback.

7. *The studies noted above have already established that some of the "downstream" enzymes reported in the current study have activity on halogenated tryptophans. For example, O'Connor established that TDCs are active on halogenated Trp and that the resulting tryptamines can undergo subsequent incorporation into alkaloids (see the second "if" in (1) above). Reference 68 shows that iaaM has activity on various halogenated Trp derivatives. This study should probably*

be cited earlier as it describes extensive studies on in vivo halogenation of fed Trp derivatives (this study also came out in 2021, so I don't think it should be described as "very recent"). Reference 73 notes that McbB tolerates fluorinated Trp; same for TsrM in reference 80. Studies noted in (1) have shown that other enzymes are also compatible with halogenated Trp derivatives. The authors need to clearly indicate that this general goal is well precedented and carefully review the literature to indicate when compatibility with halogenated Trp derivatives has already been established. The discussion of these enzymes should also reflect the precedent established by these previous reports. Lines like "we show that many enzymes are amenable to further convert..." should be changed to indicate that this is consistent with earlier reports.

We thank the reviewer for this comment. Where applicable, we have included in the text relevant prior works related to the precedent of conversion of halogenated tryptophan. These include: "RgnTDC and iaaM are thus remarkably promiscuous enzymes, evidenced by their ability to convert all positions of halogenated tryptophan, supported by similar reports in prior literature^{70,71}. All possible molecules 4b, 4c, 4d, 4e, 4f, and 4g were also observed when TnaA was expressed, thus echoing similar trends of promiscuity reports from this enzyme⁵³." In addition, for McbB, we have clarified: "Previous reports have observed McbB's ability to convert certain fluorine-modified tryptophans⁷⁶, though this represents the first confirmation of larger halogen substituted tryptophans. The four observed molecules are new-to-nature and represent the first instance of chloro- and bromo-modified beta- carbolines, with the 7-position highlighted in Fig. 6B." For TsrM, a clause to clarify this is already present "...reflecting similar trends seen in vitro⁸³."

As for the discussion, we have incorporated the clause "...largely consistent with previous reports of promiscuity." to clarify these claims

The reviewer's comment has been adequately addressed in the point-by-point reply.

Response: We thank the reviewer for this feedback.

8. Some measure of conversion needs to be provided for the downstream enzyme products. Ideally, an internal standard would have been used, but if not, the authors could report the ratio of the halogenated product to Trp and the halogenated Trp from the mass spectra acquired. In general, it is important to give some indication of how complex the product mixtures are because isolating these compounds, particularly halogenated isomers, can be challenging. Currently, the authors only show extracted ion chromatograms, which show even trace amounts of the desired compound and give a limited sense of product purity. The ion counts on these range from 10^4 - 10^6 even within a given product class, suggesting large differences in yield. Even given that limitation, product isomers are also shown in some traces, like Fig. S25, S36-S42, S48, S68-S69, S72-S78, etc. This suggests that the halogenases are active on the products formed since the selectivity on tryptophan itself is very high. If this is the case, I suggest the authors also look for doubly halogenated products. My group and others have shown that substrates similar to some of those used (e.g. tryptamine) can undergo dihalogenation. Formation of such mixtures significantly compromises the goal of producing many different compounds, so this must be clarified.

We thank the reviewer for this comment. Though we agree that conversion is a valuable metric for further optimization, we feel it is outside the scope for the current study. In this work (which we feel is rather extensive at this moment), our purpose here was to demonstrate access to a wide range of halogenated molecules in a de novo fashion from sugar via the coupling of metabolic

engineering and synthetic biology tools with synthetic chemistry approaches. Each subsequent reaction demonstrated in this work could be optimized to a further extent in future works, with time and resources entirely dedicated to quantification of yield, product purity, etc. It should be evident enough from the ion counts the relative amount of each product formed, which can form the basis to inspire future work on the topic to the broad audience of Nature Communications. Furthermore, additional analysis of the LC-MS data did not reveal any evidence of di-halogenation. We have incorporated a sentence into the text to make it clear to the readers: “The potential for di-halogenation of these downstream molecules was also investigated and was not evident based on the resulting LC-MS data.”

Reviewer’s comment not adequately addressed: *I strongly agree with the reviewers comment that a measure of conversion needs to be provided for the enzymatic products reported in the manuscript. This has clearly not been taken into consideration by the authors, who indicate that the quantification of yields will be considered in future work. For the level of impact of this journal I don’t this is a satisfactory answer.*

Response: We thank the reviewer for this comment. While we are unfortunately unable to obtain standards for the majority of the downstream halogenated products, we have now provided metrics for estimated feeding assay and coculture titers (Supplementary Table 2), feeding assay conversion values (Supplementary Table 3), and coculture yields (Supplementary Table 4) to give readers a comparison between different enzymes and halogenation positions investigated in this study. For the feeding assays, estimated titers were calculated based on amount of fed precursor consumed by the culture (As each assay was fed a fixed 0.5 mM of precursor). Implicit in this assumption is that all consumed precursor went towards formation of the intended product. Estimated titers of cocultures were taken by dividing the LCMS abundance value for the product of interest by the abundance value of that same product from the corresponding feeding assay, to avoid any biases which would have arisen by dividing LCMS abundance values of different molecules (due to the potential for differences in molecule ionization). That fraction was multiplied by the feeding assay estimated titer to obtain a value for the coculture estimated titer. Feeding assay conversion percent estimates were obtained by dividing feeding assay estimated titers by amount of fed precursor (using a millimolar basis) and multiplying by 100%. Coculture yield values (specifically looking at yield from produced tryptophan or halo-tryptophan precursor) were obtained by calculating the residual tryptophan and/or halo-tryptophan in each coculture reaction and dividing the estimated titer by the sum of the estimated downstream product titer + residual tryptophan + residual halo-tryptophan, all using a millimolar basis. We have referred to all these added metrics in the text and feel that this information can provide a sufficient estimates for titer, conversion, and yield at this time in the absence of analytical standards to obtain more precise measurements.

9. The final section on combining halogenases with downstream elaboration needs to better cite the many examples noted above that explored the same idea, including the chemoenzymatic approach of Goss. Also, the authors should note that the enzyme they ultimately used in this regard is the same one that O'Connor used in planta.

We thank the reviewer for this comment. Many of these references and points of discussion are included in the introduction of the text which sets the stage for the rest of the paper. In addition, we have included the clause to clarify that this idea has been explored in the previous works

“...which represents a break from the convention of previous downstream diversification of halogenated tryptophan works focused on combined pathway engineering within single cells 42,43,70” at the beginning of the section on page 13. Regarding the second comment, the downstream enzyme used by O’Connor in planta was a tryptophan decarboxylase from C. roseus, whereas we utilized a tryptophan decarboxylase from Ruminococcus gnavus. If you are referring to the halogenase utilized, RebH is used in both studies. We have clarified this point in the text on page 7 “...which was previously used for other de novo halogenated molecule production in planta43.

The reviewer’s comment has been adequately addressed in the point-by-point reply.

Response: We thank the reviewer for this feedback.

10. On the final paragraph of page 3, the authors list a few different flavin- and Fe(II)/alpha KG-dependent halogenases. If the authors wish to keep the discussion of the latter, they should clarify that these are different classes of enzymes and discuss them separately. For example, the "family of radical amino acid halogenases" is from the same family as WelO5, and all of these are relatively recent additions to known Fe(II)/alpha KG-dependent halogenases. The authors should be sure to cite studies on originally characterized family members like SyrB2. Alternatively, they could just focus on flavin dependent halogenases since those are the focus of this study.

We thank the reviewer for this comment. We have updated the text to better reflect the different classes of halogenases, also including the haloperoxidases and SAM-dependent halogenases. The text has been updated as follows: “In nature, halogenase enzymes generate precisely halogenated end products through a variety of reaction mechanisms²⁷. To this end, an array of halogenases have been discovered and can be characterized into four main classes. These include, for example, members of the Fe(II)/alpha KG-dependent class of halogenases such as SyrB²²⁸, BesD and similar enzymes²⁹, and late-stage halogenase WelO⁵³⁰. Other classes consist of the haloperoxidases and SAM-dependent halogenases¹⁷. Lastly, flavin-dependent halogenases constitute the final class, including RadH³¹ and Rdc^{232,33}, late-stage halogenase MalA³⁴, and, of particular interest for this study, the tryptophan halogenases such as RebH, PyrH, and Thal^{35–37}

The reviewer’s comment has been adequately addressed in the point-by-point reply.

Response: We thank the reviewer for this feedback.

11. The sizes of the chemical structures throughout the manuscript need to be greatly increased. Text and other images in the figures are also often too small to easily read.

We thank the reviewer for this comment. Chemical structures and text have been increased where needed across all figures. In addition, Figures 5 and 7 have subsequently been split into two additional figures to yield a total of 8 main text figures to provide improved readability.

12. I think that the bottom of Figure 2 is meant to show two plasmids, but nesting them as shown is confusing. The plasmids should simply be shown side-by-side. Also, the Figure 2 caption mentions panel C instead of B.

We thank the reviewer for this comment. The outer loop represents the genome and necessary

modifications to improve tryptophan production. We've updated the figure description to include "The outer loop within the bottom schematic of (A) represents the genome and relevant genomic modifications, including deletions of TrpR and TnaA and the integration of a precursor module containing SerA(fbr) and AroG(fbr) within the rbsAR locus." In addition, the Figure 2 caption has been updated to reference B.

13. Figure 6 should be moved to the SI. It is redundant with the content of Figure 5.

We thank the reviewer for this comment. Figure 6 has been moved to the SI and is now Supplementary Fig. 4. All other figure numbers have been adjusted to reflect this change.

Reviewer 2:

Remarks to the Author:

The manuscript by Reed et al describes a nicely done study to produce halogenated tryptophan derivatives in Escherichia coli. They take advantage of a large body of literature on flavin-dependent tryptophan halogenases to select variants that can both chlorinate and brominate at three positions on the 6-membered ring of the indole (positions 5, 6, and 7) to produce 6 different tryptophan analogs in vivo. This task includes the optimization and screening of different halogenases and inclusion of a flavin reductase to develop a general procedure that yielded good conversion at 30 C. While the need for a flavin reductase and two of the halogenases have been well-characterized (RebH and ThaI), the authors report a new homolog (XsHa1) that behaved well for production of 5-Cl and 5-Br Trp. It was also nice to see that the authors optimized production of brominated analogs such that formation of chlorinated side-product was limited to <10% of the final product yields. In addition to this work, they report the engineering of a strain capable of producing 300-700 mg/L of halogenated tryptophans by deleting tryptophan degradation pathways as well as overexpressing versions of the Trp operon and syrA-aroG (precursor formation) that have the known feedback inhibition mechanisms modified. The authors then identified different downstream enzymes for further transformation of Trp via different Trp metabolic pathways to screen. Screening of either spent media or co-culture showed that a number of transformations were possible to generate a good range of Trp derivatives.

Overall, I find the study to be complete and well executed with a nice range of supporting information showing screening and optimization results as well as docking studies. I also think that an interesting set of compounds were made. However, I have a major concern with the novelty of the study as there is a significant precedence of the enzymatic production of these types of halogenated tryptophans as well as their ability to be processed downstream by other metabolic and biosynthetic pathways. So overall, while I think that this study is clearly the most comprehensive and is also improved by the inclusion of Trp metabolic engineering for in vivo production, I regret to say that I do not believe that it reaches the level of impact and novelty needed for publication in this journal.

We thank the reviewer for these overall comments. We are happy to see that that this reviewer recognized the breadth and comprehensiveness of this study with respect to number of products produced. We hope that these revisions are now suited for this journal with respect to impact and novelty.

Some minor comments: - The downstream conversion is an important aspect of the manuscript but

I wasn't sure how the yield quantification was being reported. It appears to be just detected or not detected but actual yield should be established.

We thank the reviewer for this comment. Though we agree that yield quantification is a valuable metric for further optimization, we feel it is outside the scope for the current study. The purpose of this study was to demonstrate access to a wide range of halogenated molecules in a de novo fashion from sugar via the coupling of metabolic engineering and synthetic biology tools with synthetic chemistry approaches. Each subsequent reaction demonstrated in this work could be optimized to a further extent in future works, with time and resources entirely dedicated to quantification of yield, product purity, etc. It should be evident enough from the ion counts the relative amount of each product formed, which can form the basis to inspire future work on the topic to the broad audience of Nature Communications.

Reviewers comment not adequately addressed: *I would agree with the reviewers comment that yield quantification has not been adequately addressed within the manuscript. While indeed extracted ion chromatograms have been provided for each product formed, for the calibre of this journal, conversions need to be reported. An example of isolated product yields would also be a valuable addition to gauge the utility of this system while also acting as a benchmark for future optimizations.*

Response: We thank the reviewer for this comment. As detailed above, while we are unfortunately unable to obtain analytical standards for the majority of the downstream halogenated products, we have now provided metrics for estimated feeding assay and coculture titers (Supplementary Table 2), feeding assay conversion values (Supplementary Table 3), and coculture yields (Supplementary Table 4) to give readers a comparison between different enzymes and halogenation positions investigated in this study. The details for all these metrics are provided above in the response to Reviewer 1's comment 8. We have referred to all these added metrics in the text and feel that this is a sufficient representation of estimates for titer, conversion, and yield, as we do not have the analytical standards to obtain more precise measurements. We have provided yield estimates for the coculture products to show the degree of purity compared to residual halo-tryptophan or tryptophan and feel this is a valuable representation to facilitate future optimization.

- I find that some of the figures are a bit hard to parse when the yields are shown as heat maps (such as in Figure 1C) and the structures are a bit hard to see in the latter figures on downstream conversion (Figures 5 and 7). Maybe Figure 6 can be incorporated into Figure 7 some way since having more clarity into the structures would be more useful than the schematic aspect since many of the concepts are incorporated into earlier figures.

We thank the reviewer for this comment. We have moved Figure 6 to the SI and it is now Supplementary Fig. 4. Figures 5 and 7 have subsequently been split into two additional figures to yield a total of 8 main text figures. We believe that the heat map for Figure 1C is an acceptable representation of the data and especially highlights the effectiveness of halogenase XsHal, along with various other halogenases at certain temperatures.

Reviewers comment partially addressed: *I would agree with the reviewer that a scale which also represents % L-Trp conversion needs to be included in Figure 1C. While the representation of the data at present highlights the effectiveness of the tested halogenases at certain temperatures it*

does not accurately represent conversion.

Response: We thank the reviewer for this feedback. The scale of the heat map for Figure 1C is in terms of % L-Trp conversion. To make this clearer, we have labeled each panel in the heat map with the corresponding numerical conversion value.

Reviewers' Comments:

Reviewer #3:

Remarks to the Author:

See attached pdf

Response to Reviewer Comments

****Note:** Responses to reviewer comments in this round of review are listed in Purple**

Reviewer #3 (Remarks to the Author):

Due to the unavailability of the prior reviewers for the re-assessment of the manuscript by Reed et al. I was asked on my opinion whether the authors have appropriately addressed the remaining comments from the previous reviewers and on my opinion on the overall suitability of the manuscript for publication in Nature Communications.

My overall assessment of the manuscript can be found directly below. My opinion on the completeness of the author's responses to the comments of the previous reviewers can be found highlighted in red after each comment in the point-by-point reply (attached document).

Overall assessment:

Reed et al present an interesting manuscript which describes the de novo synthesis of a range of halogenated tryptophans in Escherichia coli from glucose as a starting material. In this work, the authors have leveraged both well-known (e.g., RebH and Thal) and a new flavin-dependent halogenase (XsHal) by incorporating them into an engineered strain of E.coli capable of producing 300-700 mg/L of halogenated tryptophans. They subsequently identified a set of downstream enzymes that can act on the halogenated tryptophans and report the biosynthesis of 26 distinct halogenated compounds.

Having reviewed the authors point-by-point reply to the reviewer's comments, I feel that they have only partially addressed the concerns of the reviewers. While I think the authors have taken reviewer 1's concerns regarding referencing and context into consideration by adapting the main text, they have not incorporated a measure of conversion or yield for the downstream enzymatic products reported. Furthermore, the products have not been characterized or compared to authentic reference standards. This makes it very difficult to benchmark their system. I would also agree with the overall impression of the previous reviewers, that - given the wealth of literature which is available on the enzymatic halogenation of tryptophans, in addition to their ability to be further modified by downstream enzymes – this manuscript lacks the novelty required for publication in Nature Communications.

Response: We thank the reviewer for this overview and review of the manuscript. To further address the prior reviewers' comments, we have incorporated measurements of conversion and yield for appropriate experiments into the supplementary information and provide some additional details listed in the response to the specific point-by-point comments below. It should be noted that we are unfortunately unable to access authentic reference standards for the vast majority of downstream halogenated tryptophan products made de novo in this work and thus cannot provide exact titers. As a result, we have provided estimates for titers, conversions, and yields using assumptions based on amount of fed precursor consumed, as well as comparisons of LCMS abundance for molecules in the feeding assays and coculture experiments. Full details for these assumptions and calculations are listed below in this document (in the updated response to Reviewer 1's Comment 8) as well as in the revised manuscript. As previously discussed, while prior works have focused on specific tryptophan halogenation reactions or downstream modifications, we believe this to be the first and most comprehensive work fully showcasing the combination of metabolic engineering, coculture design, and combinatorial chemistry to access a wide array (26 distinct downstream products) of halo-tryptophan-derived compounds in a purely de novo fashion from simple sugar starting materials. We thus feel the study is novel compared to prior works.

While the authors have improved on the manuscript by incorporating measurements of conversion and yield for their feeding assays and co-culture experiments, I have concerns regarding how these metrics were calculated. It's understandable that authentic reference standards are not always available, but I think that these calculations should at least be benchmarked using reference standards that are commercially available (for example, 5-Cl-indole or 6-Br-indole). This would provide a more accurate sense of the product yields reported.

Additional comments to figures and supplementary figures:

- Line 189: Table 2 cited in the text is not provided

Response: We thank the reviewer for this comment. Table 2 has been provided in the revised version of the manuscript.

My comment has been adequately addressed by the authors.

- Line 225: It would be useful to reiterate the figure no. when referring to “shown in white in the Product Observed column”

Response: We thank the reviewer for this comment. We have updated the text to include a reference to Figure 3b.

My comment has been adequately addressed by the authors.

- Supplementary figures 5 and 6: It would be useful to label residues shown in docking studies – especially the Tyr216 which undergoes a conformational change in McbB.

Response: We thank the reviewer for this comment. We have labeled the Tyr216 in Supplemental Figures 5 and 6, as well as denoted whether it is in the up or down conformation. This should add clarity to the reader.

My comment has been adequately addressed by the authors.

Additional Comments:

Line 263: Please correct to “While we were not able to obtain analytical standards for the majority of downstream halogenated molecules, we have provided estimated titers and conversions of fed substrates from the feeding assay based on consumption of fed substrate.”

Methods Section: The degree symbol is missing throughout the methods section when temperature is referred to.

Response to Reviewer Comments

****Note:** Responses to reviewer comments in this round of review are listed in **Orange** and all older text is now italicized**

Reviewer #3 (Remarks to the Author):

Due to the unavailability of the prior reviewers for the re-assessment of the manuscript by Reed et al. I was asked on my opinion whether the authors have appropriately addressed the remaining comments from the previous reviewers and on my opinion on the overall suitability of the manuscript for publication in Nature Communications.

My overall assessment of the manuscript can be found directly below. My opinion on the completeness of the author's responses to the comments of the previous reviewers can be found highlighted in red after each comment in the point-by-point reply (attached document).

Overall assessment:

Reed et al present an interesting manuscript which describes the de novo synthesis of a range of halogenated tryptophans in Escherichia coli from glucose as a starting material. In this work, the authors have leveraged both well-known (e.g., RebH and Thal) and a new flavin-dependent halogenase (XsHal) by incorporating them into an engineered strain of E.coli capable of producing 300-700 mg/L of halogenated tryptophans. They subsequently identified a set of downstream enzymes that can act on the halogenated tryptophans and report the biosynthesis of 26 distinct halogenated compounds.

Having reviewed the authors point-by-point reply to the reviewer's comments, I feel that they have only partially addressed the concerns of the reviewers. While I think the authors have taken reviewer 1's concerns regarding referencing and context into consideration by adapting the main text, they have not incorporated a measure of conversion or yield for the downstream enzymatic products reported. Furthermore, the products have not been characterized or compared to authentic reference standards. This makes it very difficult to benchmark their system. I would also agree with the overall impression of the previous reviewers, that - given the wealth of literature which is available on the enzymatic halogenation of tryptophans, in addition to their ability to be further modified by downstream enzymes – this manuscript lacks the novelty required for publication in Nature Communications.

***Response:** We thank the reviewer for this overview and review of the manuscript. To further address the prior reviewers' comments, we have incorporated measurements of conversion and yield for appropriate experiments into the supplementary information and provide some additional details listed in the response to the specific point-by-point comments below. It should be noted that we are unfortunately unable to access authentic reference standards for the vast majority of downstream halogenated tryptophan products made de novo in this work and thus cannot provide exact titers. As a result, we have provided estimates for titers, conversions, and yields using assumptions based on amount of fed precursor consumed, as well as comparisons of LCMS abundance for molecules in the feeding assays and coculture experiments. Full details for these assumptions and calculations are listed below in this document (in the updated response to Reviewer 1's Comment 8) as well as in the revised manuscript. As previously discussed, while prior works have focused on specific tryptophan halogenation reactions or downstream modifications, we believe this to be the first and most comprehensive work fully showcasing the combination of metabolic engineering, coculture design, and combinatorial chemistry to access a wide array (26*

distinct downstream products) of halo- tryptophan-derived compounds in a purely de novo fashion from simple sugar starting materials. We thus feel the study is novel compared to prior works.

While the authors have improved on the manuscript by incorporating measurements of conversion and yield for their feeding assays and co-culture experiments, I have concerns regarding how these metrics were calculated. It's understandable that authentic reference standards are not always available, but I think that these calculations should at least be benchmarked using reference standards that are commercially available (for example, 5-Cl-indole or 6-Br-indole). This would provide a more accurate sense of the product yields reported.

Response: We thank the reviewer for this comment. We have added a **Supplementary Table 5** which benchmarks estimated coculture titers of 5-Cl-Tryptamine and 5-Br-Tryptamine against those quantified via an analytical standard. The estimated titers show fairly good agreement with the exact titers and we have added a sentence to the main text to highlight this addition: "These estimated titers have been benchmarked against exact titers for 5-Cl-Tryptamine and 6-Cl-Tryptamine, quantified via use of commercially available analytical standards (**Supplementary Table 5**), demonstrating good agreement from an order of magnitude standpoint." As these estimated coculture titers were calculated using both feeding assay precursor consumption and LCMS abundances (the full suite of metrics used for estimated calculations in Supplementary tables 2-4) this adds validity to our set of assumptions used in estimating these metrics. Of course, the reported titers, yields, and conversions in Supplementary Tables 2-4 are all clearly stated to be estimates and all assumptions used in their calculations have been clearly stated in the main and supplementary text. We feel that this is a very comprehensive assessment of the approximate titers and purity of the reported compounds and is the closest we can get to full quantification due to our inability to obtain analytical standards for the majority of molecules produced.

Additional comments to figures and supplementary figures:

- Line 189: Table 2 cited in the text is not provided

Response: *We thank the reviewer for this comment. Table 2 has been provided in the revised version of the manuscript.*

My comment has been adequately addressed by the authors.

Response: We thank the reviewer for this feedback.

- Line 225: It would be useful to reiterate the figure no. when referring to "shown in white in the Product Observed column"

Response: *We thank the reviewer for this comment. We have updated the text to include a reference to Figure 3b.*

My comment has been adequately addressed by the authors.

Response: We thank the reviewer for this feedback.

- Supplementary figures 5 and 6: It would be useful to label residues shown in docking studies especially the Tyr216 which undergoes a conformational change in McbB.

Response: *We thank the reviewer for this comment. We have labeled the Tyr216 in Supplemental*

Figures 5 and 6, as well as denoted whether it is in the up or down conformation. This should add clarity to the reader.

My comment has been adequately addressed by the authors.

Response: We thank the reviewer for this feedback.

Additional Comments:

Line 263: Please correct to “While we were not able to obtain analytical standards for the majority of downstream halogenated molecules, we have provided estimated titers and conversions of fed substrates from the feeding assay based on consumption of fed substrate.”

Response: We thank the reviewer for this comment. The sentence has been corrected in the main text.

Methods Section: The degree symbol is missing throughout the methods section when temperature is referred to.

Response: We thank the reviewer for this comment. The degree symbol has been added throughout the text for temperature references.

Reviewers' Comments:

Reviewer #3:

Remarks to the Author:

Review report attached.

A Modular, Synthetic Biosynthesis Platform for de novo Production of Diverse Halogenated Tryptophan-Derived Molecules

Reviewer #3 (Remarks to the Author):

While the authors have improved on the manuscript by incorporating measurements of conversion and yield for their feeding assays and co-culture experiments, I have concerns regarding how these metrics were calculated. It's understandable that authentic reference standards are not always available, but I think that these calculations should at least be benchmarked using reference standards that are commercially available (for example, 5-Cl-indole or 6-Br-indole). This would provide a more accurate sense of the product yields reported.

Author's Response: We thank the reviewer for this comment. We have added a **Supplementary Table 5** which benchmarks estimated coculture titers of 5-Cl-Tryptamine and 5-Br-Tryptamine against those quantified via an analytical standard. The estimated titers show fairly good agreement with the exact titers and we have added a sentence to the main text to highlight this addition: "These estimated titers have been benchmarked against exact titers for 5-Cl-Tryptamine and 6-Cl-Tryptamine, quantified via use of commercially available analytical standards (**Supplementary Table 5**), demonstrating good agreement from an order of magnitude standpoint." As these estimated coculture titers were calculated using both feeding assay precursor consumption and LCMS abundances (the full suite of metrics used for estimated calculations in Supplementary tables 2-4) this adds validity to our set of assumptions used in estimating these metrics. Of course, the reported titers, yields, and conversions in Supplementary Tables 2-4 are all clearly stated to be estimates and all assumptions used in their calculations have been clearly stated in the main and supplementary text. We feel that this is a very comprehensive assessment of the approximate titers and purity of the reported compounds and is the closest we can get to full quantification due to our inability to obtain analytical standards for the majority of molecules produced.

In response to my concerns regarding how metrics for co-culture experiments were calculated, the authors have compared their estimated titre values for 5-Cl-Tryptamine and 5-Br-Tryptamine with titres which were quantified using authentic reference standards (**Supplementary Table 5**). However, in the main text the authors refer to the quantification of 6-Cl-Tryptamine, which I believe to be an error based on the approximate co-culture titres reported for this compound (**Supplementary Table 2**). The authors should address this error (main text, line 324). In addition, while the quantified titre value of 5-Br-Tryptamine is in agreement with the reported estimated product titre, I find it difficult to agree that the same applies in the case of 5-Cl-Tryptamine. A discrepancy of $\geq 30\%$ is observed here. I therefore think it's somewhat misleading, when we are discussing <100 milligram scale production, to state that "this demonstrates good agreement from an order of magnitude standpoint".

A Modular, Synthetic Biosynthesis Platform for de novo Production of Diverse Halogenated Tryptophan-Derived Molecules

Reviewer #3 (Remarks to the Author):

While the authors have improved on the manuscript by incorporating measurements of conversion and yield for their feeding assays and co-culture experiments, I have concerns regarding how these metrics were calculated. It's understandable that authentic reference standards are not always available, but I think that these calculations should at least be benchmarked using reference standards that are commercially available (for example, 5-Cl-indole or 6-Br-indole). This would provide a more accurate sense of the product yields reported.

Author's Response: *We thank the reviewer for this comment. We have added a **Supplementary Table 5** which benchmarks estimated coculture titers of 5-Cl-Tryptamine and 5-Br-Tryptamine against those quantified via an analytical standard. The estimated titers show fairly good agreement with the exact titers and we have added a sentence to the main text to highlight this addition: "These estimated titers have been benchmarked against exact titers for 5-Cl-Tryptamine and 6-Cl-Tryptamine, quantified via use of commercially available analytical standards (**Supplementary Table 5**), demonstrating good agreement from an order of magnitude standpoint." As these estimated coculture titers were calculated using both feeding assay precursor consumption and LCMS abundances (the full suite of metrics used for estimated calculations in Supplementary tables 2-4) this adds validity to our set of assumptions used in estimating these metrics. Of course, the reported titers, yields, and conversions in Supplementary Tables 2-4 are all clearly stated to be estimates and all assumptions used in their calculations have been clearly stated in the main and supplementary text. We feel that this is a very comprehensive assessment of the approximate titers and purity of the reported compounds and is the closest we can get to full quantification due to our inability to obtain analytical standards for the majority of molecules produced.*

*In response to my concerns regarding how metrics for co-culture experiments were calculated, the authors have compared their estimated titre values for 5-Cl-Tryptamine and 5-Br-Trypramine with titres which were quantified using authentic reference standards (**Supplementary Table 5**). However, in the main text the authors refer to the quantification of 6-Cl-Tryptamine, which I believe to be an error based on the approximate co-culture titres reported for this compound (**Supplementary Table 2**). The authors should address this error (main text, line 324). In addition, while the quantified titre value of 5-Br-Trypramine is in agreement with the reported estimated product titre, I find it difficult to agree that the same applies in the case of 5-Cl-Tryptamine. A discrepancy of $\geq 30\%$ is observed here. I therefore think it's somewhat misleading, when we are discussing <100 milligram scale production, to state that "this demonstrates good agreement from an order of magnitude standpoint".*

Response: We thank the reviewer for this comment. We have corrected this typo in the main text to state that "estimated titers have been benchmarked against exact titers for 5-Cl-Tryptamine and 5-Br-Tryptamine, quantified via use of commercially available analytical standards (**Supplementary Table 5**)". To alleviate the reviewer's concern about potential for misleading text, we have amended our discussion of these results to state "The estimated 5-Br-Tryptamine titer falls within 5% of the exact titer, while the estimated 5-Cl-Tryptamine titer exceeds the exact titer by ~50%. This indicates that the estimation accuracy likely varies with each compound. However, these estimates provide a general sense of titer scale (e.g. 1 mg/L vs 10 mg/L vs 100 mg/L), providing our best approximation in the absence of available reference standards for the majority of downstream halogenated compounds produced in this study." As stated in the amended text, the provided estimates give the readers a sense of scale (in terms of full orders of magnitude such as 1 vs 10 vs 100 vs 1000 mg/L) in the absence of availability of reference standards. **We have not claimed** in the text that these estimated titers are exact **nor have we claimed to have achieved maximum titers**. We feel that this is the best that one can do here in the absence of having an authentic reference standard for each compound.

Reviewers' Comments:

Reviewer #3:

Remarks to the Author:

I'm satisfied with the updates/ explanations made on yields.